# Comparative Analysis of Chemical Profile and Biological Activity of *Juniperus communis* L. Berry Extracts

**DOI:** 10.3390/plants12193401

**Published:** 2023-09-27

**Authors:** Timur Belov, Dmitriy Terenzhev, Kseniya Nikolaevna Bushmeleva, Lilia Davydova, Konstantin Burkin, Igor Fitsev, Alsu Gatiyatullina, Anastasia Egorova, Evgeniy Nikitin

**Affiliations:** 1Arbuzov Institute of Organic and Physical Chemistry, FRC Kazan Scientific Center of RAS, Arbuzov Str. 8, 420088 Kazan, Russia; dmitriy.terenzhev@mail.ru (D.T.); ks.bushmelewa09@yandex.ru (K.N.B.); oleym73@inbox.ru (L.D.); berkutru@mail.ru (E.N.); 2Federal State Budgetary Scientific Institution, Federal Center for Toxicological, Radiation, and Biological Safety, Nauchny Gorodok-2, 420075 Kazan, Russia; konstantinburkin@yandex.ru; 3A.M. Butlerov Chemical Institute, Kazan Federal University, Kremlevskaya Str. 18, 420008 Kazan, Russia; fitzev@mail.ru; 4Research Institute for Problems of Ecology and Mineral Wealth Use of Tatarstan Academy of Sciences, IPEM TAS, Daurskaya Str. 28, 420087 Kazan, Russia; tbkalinnikova@gmail.com (A.G.); egorovanastassia@gmail.com (A.E.)

**Keywords:** *Juniperus communis* L., berries, extract, GC-MS, antimicrobial activity, antioxidant activity, nematocidal activity

## Abstract

Researchers are looking for the most effective ways to extract the bioactive substances of *Juniperus communis* L. berries, which are capable of displaying the greatest range of biological activity, namely antimicrobial potential “against phytopathogens”, antioxidant activity and nematocidal activity. This study provides detailed information on the chemical activity, group composition and biological activity of the extracts of juniper berries of 1- and 2-year maturity (JB1 and JB2), which were obtained by using different solvents (pentane, chloroform, acetone, methanol and 70% ethanol) under various extraction conditions (maceration and ultrasound-assisted maceration (US)). Seventy percent ethanol and acetone extracts of juniper berries were analyzed via gas chromatography–mass spectrometry, and they contained monoterpenes, sesquiterpenes, polysaccharides, steroids, fatty acid esters and bicyclic monoterpenes. The antimicrobial activity was higher in the berries of 1-year maturity, while the acetone extract obtained via ultrasound-assisted maceration was the most bioactive in relation to the phytopathogens. Depending on the extraction method and the choice of solvent, the antioxidant activity with the use of US decreased by 1.5–1.9 times compared to the extracts obtained via maceration. An analysis of the nematocidal activity showed that the sensitivity to the action of extracts in *Caenorhabditis elegans* was significantly higher than in *Caenorhabditis briggsae*, particularly for the acetone extract obtained from the juniper berries of 1-year maturity.

## 1. Introduction

The demand for medicinal products and biologically active additives of plant origin increases every year. The identification of biologically active substances in natural raw materials is one of the priority areas of chemistry, biochemistry and pharmacy [1]. The content of biological active substances in plants depends on many factors, including taxonomy, altitude, climatic conditions, soil type, soil composition, etc., which cause changes in the qualitative and quantitative contents of biological compounds in different regions of the planet [2,3].

The genus Juniperus is one of the most diverse among conifers and includes more than 60 species. Juniper (*Juniperus communis* L.) is an evergreen shrub of the genus *Juniperus* of the Cypress family, growing in the forests of Russia and forest-steppe zones of parts of Europe, including western and part of eastern Siberia. Its population is spread globally, being the only Juniperus species that is found in both hemispheres, with reports of this plant in the Arctic regions of Asia and North America [4]. These are dicotyledonous trees that produce seeds every 2 years, which have a circular or spherical shape [5]. In Tatarstan, it is the most widespread on the territory of the Volga-Kama State Natural Biosphere Reserve (55°18′10″ N 49°17′10″ E) and the territories adjacent to it [6,7].

It has been established that the common juniper berries contain a wide range of biologically active compounds: α-pinene, camphene, pectin, organic acids (glycolic, malic and ascorbic), cyclohexitol, terpenes, proteins, fermentable sugars, wax, gum, cadinene and juniper camphor [8]. To date, several reports have highlighted their antimicrobial, antifungal, antioxidant, anti-inflammatory and antidiabetic potentials, as well as their anticarcinogenic, hepatoprotective, neuronal and renal effects [2,9,10]. Due to this, essential oils, extracts, biologically active fractions and individual compounds from juniper berries can be useful for the development of new pharmacological drugs in the treatment of a number of acute and chronic human diseases.

All juniper species are characterized by a high content of essential oils and phenolic compounds and are largely used in folk medicine in various countries, showing a wide range of biological activity and industrial applications [11]. It is known that juniper has diuretic, anti-inflammatory, antifungal, analgesic, hepatoprotective, antimicrobial and antioxidant effects in traditional medicine [12].

The chemical composition and biological activity of juniper berries of 1- and 2-year maturity (JB1 and JB2), growing in the temperate continental zone of the Republic of Tatarstan, have been little studied to date [1,13], while their extracts may contain elevated concentrations of physiologically active substances and may exhibit high levels of antioxidant and antimicrobial activities. It is noteworthy that the extracts of berries of 1-year maturity still remain poorly studied from a pharmacological point of view [5,14,15].

Since juniper berries contain a wide range of compounds of various chemical nature, it is advisable to use various polar and non-polar solvents to extract the maximum possible number of components. According to the data in the literature, the most commonly used solvents are water, methanol, ethanol, butanol, ethyl acetate, pentane, chloroform, dichloromethane or a mixture thereof [16].

The trend of recent decades around the world is to reduce the use of chemical plant protection products and to pay attention to the development of phytoprotective agents based on the beneficial properties of wildlife objects: preparations based on essential oils (EOs) or plant extracts with antimicrobial properties, environmentally friendly and biodegradable properties, the ability to form non-toxic metabolites, and the ability to meet the requirements of organic farming [17]. Phytopathogens are one of the most important biological factors that pose a threat mainly to cultivated plants and, as a consequence, cause a significant decrease in yield [18]. The available information on the biological activity of JB1 and JB2 extracts with respect to phytopathogenic bacteria and fungi is scarce [19,20], unlike essential oils [21,22]. In addition, berry components may have high nematocidal activity, which is a direct correlation indicator in the development of new anthelmintic drugs [23,24].

The objective of this study is to evaluate the chemical composition and bioactivity of the extracts of juniper berries (*J. communis*) of 1- and 2-year maturity collected in the territory of the Republic of Tatarstan using various solvents (pentane, chloroform, acetone, methanol and 70% ethanol) and extraction methods—such as maceration and ultrasound-assisted maceration—for an efficient exploitation as a source of biologically active substances for pharmacological preparations and/or plant protection products. This work is designed to fill in the lack of data on the component composition of berry extracts of *Juniperus communis* during the first and second year of maturity that grow in the territory of the Republic of Tatarstan, as well as their multidirectional biological activities (antibacterial, antioxidant and nematocidal activities) [1,2,15,20,25].

## 2. Results

### 2.1. Physico-Chemical Properties

The extraction yield, pH and electrical conductivity data are shown in Table 1.

The pH values of the JB1 extracts varied depending on the choice of solvent and ranged from 4.97 to 7.15. In the case of the JB2 extracts, the pH ranged from 5.76 to 6.79.

The electrical conductivity of JB1 extracts ranged from 10.17 to 57.35 mS/cm; for JB2 extracts the results were significantly lower—from 7.55 to 35.55 mS/cm. The choice of solvent and extraction method also affected the extraction yield of the extract itself. For JB1 extracts, it ranged from 1.38 to 7.33%, with the maximum value obtained using 70% ethanol. For JB2 extracts, concentrations ranged from 0.63 to 8.79%.

### 2.2. Quantification of The Main Phytochemicals

The results of the total sugar, phenolic, flavonoid, terpenoid and ascorbic acid contents are presented in Table 2. The total amount of flavonoids and phenolics was maximum for JB1 extracts. According to the table, with the increasing polarity of the solvent, the amount of extracted phenolics and flavonoids in JB1 extracts differed from that of JB2 extracts, with the maximum difference being more than four times when using 70% ethanol extract with maceration.

As shown in Table 2, the highest terpenoid content corresponded to the JB1 extracts and was ranked according to the choice of solvent: 70% ethanol > methanol > acetone > chloroform > pentane. The use of ultrasound led to a decrease in the total content of terpenoids, while in JB1, the terpenoid content was more than 1.5–2 times higher than in JB2 extracts, which can be characterized as a decrease in the secondary metabolites produced by juniper berries during their maturation. 

The ascorbic acid content of JB1 and JB2 extracts using 70% ethanol showed high values in a number of solvents. At the same time, the ascorbic acid content of JB1 and JB2 extracts using pentane and chloroform during maceration and ultrasound-assisted maceration was below the limit of detection. The concentration of ascorbic acid depended on the degree of berry ripening; when methanol, acetone and 70% ethanol extracts were used, it was exceeded by more than 3–8 times for JB1 compared to JB2 extracts.

The total sugar content in the extracts studied was higher for JB2 extracts. From the data in Table 2, it can be concluded that the solvents can be ranked according to the sugar content as follows: 70% ethanol > acetone > methanol > chloroform. The total sugar content in JB1 and JB2 pentane extracts during maceration and ultrasound-assisted maceration and in JB1 chloroform extracts was not determined. The total sugar content in JB1 extracts was higher when using ultrasound (21.39 mg/g dry extract), which was 2.5 times lower for a similar extract from JB2 (53.01 mg/g dry extract).

#### Statistical Analysis of Influence of Factors on Extraction Yield and the Main Phytochemicals

Statistical analysis was performed using the Post-hoc Tukey test and is presented in Table 3. A significant effect of maturity year, extraction method and solvent on extract yield and main phytochemicals (total sugars, total phenolic compounds, total flavonoids, total terpenoids and ascorbic acid content) was identified (*p* = 0.0). Multiple analysis of variance (MANOVA) was performed to analyze the effect of extraction methods and is included as Appendix A (Appendix A).

### 2.3. GS-MS Analysis

The results of the research, using the GC-MS method with different ionization methods on samples of acetone and 70% ethanol extracts obtained from JB1 and JB2, identified a number of major and minor components (Table 4 and Appendix A), significantly extending and clarifying the list thanks to previously known studies. These extracts showed a higher biological profile (antioxidant and bacterial activity) and were selected for further study. The chromatograms of the tested samples are included as a Appendix A (Appendix A). A total of 38 individual compounds were identified depending on the selected extractant, while a total of 23 compounds were identified for JB1 extracts and 24 compounds for JB2 extracts. Their identification was performed by the relative matching factors (RMFs) generated by NIST MS Search Program software during the automatic deconvolution of the experimental mass spectrum and their comparison with those available in electronic reference libraries [26].

It was found that the chemical components of the volatile fractions varied between berries of different ripening periods, and the percentage of each component also varied greatly.

The samples of JB1 extracted with 70% ethanol (JB1E), according to GC-MS, contained 20 components distributed as follows: monoterpenes (55.7%), sesquiterpenes (21.8%), polysaccharides (19.2%), steroids (1.89%) and bicyclic monoterpenes (1.71%). The following minor constituents were identified: fatty acid esters (1.52%), cyclic aldehydes (0.78%), aromatic aldehydes (0.75%), monosaccharides (0.63%), cyclic monoterpenes (0.55%) and acyclic monoterpenoids (0.43%).

A total of 21 components with a high content of monoterpenes (48.43%), sesquiterpenes (21.23%), polysaccharides (12.67%), steroids (3.65%), fatty alcohols (2.67%), fatty acid esters (2.53%), cyclic aldehydes (1.9%), aromatic aldehydes (1.83%), bicyclic monoterpenes (1.54%) and monosaccharides (1.41%) were extracted from JB1 with absolute acetone (JB1A); the minor components obtained were diterpenoids (0.72%), cyclic monoterpenes (0.5%), acyclic monoterpenoids (0.47%) and bicyclic monoterpenoids (0.44%).

For JB2, the extraction with 70% ethanol (JB2E) contained 23 components. Among them were a high content of monosaccharides (40.61%), monoterpenes (24.11%), polysaccharides (11.26%), sesquiterpenes (7.12%), α-hydroxyketones (3.44%), alcohol ketones (2.43%), polyatomic alcohols (1.87%), unsaturated ketones (1.91%), keto acids (1.13%) and lactones (1.92%); the minor constituents were pyrazoles (0.83%), monoterpene alcohol (0.96%), pyrans (0.91%), lipids (0.77%), diterpenoid (0.39%) and monoterpene ketone (0.32%).

The extraction of JB2 with absolute acetone (JB2A) revealed a high content of monoterpenes (46.79%), monosaccharides (21.04%), sesquiterpenes (10.06%), polyatomic alcohols (7.34%), polysaccharides (4.77%), fatty alcohols (4.45%) and diterpenoid (3.01%); the minor components obtained were monoterpene ketone (0.87%), monoterpene alcohol (0.85%) and α-hydroxyketones (0.83%).

According to the GC-MS data, α-pinene, dihydroxyacetone, β-myrcene, caryophyllene, α-humulene, germacrene D, germacrene D-4-ol and myo-inositol were detected in all samples. In JB1, a higher content of monoterpenes and sesquiterpenes was identified, whereas in JB2, monosaccharides predominated. Secondary metabolites, especially terpene compounds [27], are a factor of specific biological activity—antimicrobial, antioxidant and nematocidal. The specific individual components obtained from JB1 extracts were sabinene; δ-3-carene; limonene; methyl citronellate; β-farnesene; epimethenediol; (1R,4aR,5S)-5-((E)-5-methoxy-3-methylpent-3-en-1-yl)-1,4a-dimethyl-6-methylenedecahydronaphthalene-1-carbaldehyde; 2-[[2-[(2-ethylcyclopropyl)methyl]cyclopropyl]methyl]- Cyclopropaneoctanoic acid; methyl ester; and (1R,4aR,5S)-5-[(E)-5-hydroxy-3-methylpent-3-enyl]-1,4a-dimethyl-6-methylidene-3,4,5,7,8,8a-hexahydro-2H-naphthalene-1-carbaldehyde. The specific individual components obtained from JB2 extracts were acetol; glyceraldehyde; 1,2,3-propanetriol, verbenol; verbenone; and isocembrol.

The qualitative and quantitative composition of the berry extracts differed depending on the polarity of the solvent. For JB1E extract, bicyclogermacrene and methyl-5,11,14-eicosatrienoate were additionally identified; in the case of the JB1A extract, bornyl acetate and dihydroabietinol were additionally identified. JB2E yielded 1-hydroxybut-3-en-2-one; 2-hydroxy-3-oxobutanal; methyl-2-oxopropanate; 2-hydroxy-2-cyclopenten-1-one; 2-hydroxy—γ-butyrolactone; 4-methyl-1-H-pyrazole-3-carboxylic acid; tetrahydro-2-H-pyran-2-methanol; and 1,2-diacetate-1,2,3-propanetriol. Extraction with acetone additionally revealed [S-(E,Z,E,E)]-3,7,11-trimethyl-14-(1-methylethyl)-1,3,6,10-cyclotetradecatetraene.

### 2.4. DPPH-Free Radical Scavenging Assay

The study of the antioxidant activity of the extracts of JB1and JB2 was carried out for the scavenging of DPPH radicals (2,2-diphenyl-1-picrylhydrazyl).

The neutralizing effects of the extracts of immature berries on DPPH radicals during maceration were in the following order: 70% ethanol > acetone > methanol > chloroform > pentane. During ultrasound-assisted maceration, the neutralizing effects were in the following order: 70% ethanol > methanol> acetone> chloroform > pentane.

According to the results of the DPPH radical scavenging analysis, the effectiveness of unripe berries as a source of antioxidants was almost two times higher during maceration (45 °C, 1.5 h) than during ultrasound (45 °C, 15 min) (Figure 1).

The activity of unripe berries was higher when 70% ethanol was used as an extractant (3.4 and 3.9 times higher than acetone and methanol for maceration and 4 and 3.1 times higher for ultrasound-assisted maceration, respectively). 

During the maceration of JB1 (JB1-m), the bioactivity of the extracts by maceration was arranged in the following order: 70% ethanol > methanol > acetone > chloroform > pentane. During ultrasound-assisted maceration, the order of solvents changed: methanol > 70% ethanol > acetone > chloroform > pentane.

In the case of using of JB2 as an object of study, the greatest ability to remove DPPH radicals was possessed by a methanol-based extract without the use of ultrasound, with a 3.1- and 2.6-times higher than extraction with ethanol and acetone, respectively.

Among the extractants used, pentane and chloroform were confirmed to be the least efficient for the extraction of bioactive compounds. A possible reason for this may be the low polarity of the solvents.

The extracts of JB1 had a more pronounced antioxidant activity compared to JB2, which may indicate a higher content of phenolic compounds and a higher total content of monoterpenes, sesquiterpenes and other biologically active components, which were the highest in the 70% ethanol, acetone and methanol extracts. The observed high levels of antioxidant activity observed when using maceration, as opposed to ultrasound, may indicate the preservation of the antiradical properties of these fractions.

#### 2.4.1. Correlation between Phytochemicals and Antioxidant Activity of Berries

Pearson correlations (*p* < 0.05 and *p* < 0.01) were used to identify correlations between phytochemicals and the antioxidant activity of the acetone, methanol and 70% ethanol extracts of JB1 and JB2.

Table 5 shows a negative correlation between the respective phenolic, terpenoid constituents, and ascorbic acid and the IC_50_ DPPH antioxidant capacity in first-year berries (acetone, maceration) (r = −0.672; −0.659 and −0.054), while the IC_50_ DPPH antioxidant capacity with acetone and US correlated significantly with the phenolic compounds (r = 0.99), which also significantly correlated with the phenolic compounds (r = 0.925) and flavonoids (r = 0.985) with the use of methanol and US.

Table 6 shows an inverse correlation for the IC_50_ DPPH antioxidant capacity with flavonoids (r = −0.944) and ascorbic acid (r = −0.947) using methanol and maceration, also with ascorbic acid (r = −0.978) using acetone and US.

#### 2.4.2. Analysis of Chemiluminescent Activity of the Studied Compounds

The analysis of the chemiluminescence activity of the studied extracts under normal conditions showed the highest degree of luminescence quenching and the duration of the latent period for the acetone extract JB1-*m*—4173 s according to TRAP value and 99% according to TAR value (Figure 2 and Figure 3). Similarly, under ultrasound-assisted maceration conditions, the 70% ethanol extract of JB1 had the greatest activity—2079 s according to TRAP value and 97.8% according to TAR. The methanol extract of JB2 had the greatest activity under normal conditions (578 s and 94.3%) with the use of ultrasound-assisted maceration, while the 70% ethanol extract had the greatest (593 s and 93.6%). According to Figure 2 and Figure 3, the extracts of JB1 extracted using ethanol, acetone and methanol contained a greater amount of strong antioxidants in their composition, while their amount decreased when using US.

### 2.5. Antimicrobial Activity

The results of the antimicrobial activity analysis are presented in Table 7.

The minimum inhibitory concentrations (MICs) for extracts of JB1 ranged from 39 to 2500 µg/mL, and for extracts of JB2 extracts, they ranged from 156 to 2500 µg/mL. The minimum bactericidal concentrations of JB1 extracts for *Clavibacter michiganensis* ranged from 39 to 312 µg/mL, and for *Erwinia carotovora* spp., they ranged from 39 to 2500 µg/mL; in the case of JB2, from 156 to 2500 µg/mL for *Clavibacter michiganensis,* and from 1250 to 2500 µg/mL for *Erwinia carotovora* spp. The minimum fungicidal concentrations for *Rhizoctonia solani* and *Alternaria solani* were 78–2500 µg/mL for extracts of JB1 and 625–2500 µg/mL for extracts of JB2.

The extracts of common juniper obtained with different extractants and conditions had different antimicrobial activity against phytopathogens. According to Table 7, it can be said that all the solvents (extractants) tested can be ranked. For JB1, acetone > 70% ethanol > chloroform > pentane > methanol; for JB2, methanol > acetone> chloroform > pentane > 70% ethanol.

The use of ultrasound on JB1 and JB2 extracts resulted in an increase in antimicrobial activity in the case of acetone, methanol and pentane, and a decrease in antimicrobial activity in the case of 70% ethanol. This may also indicate the extraction of associated compounds that are less bioactive to bacteria/fungi, and the possible mutual transformation of extractive compounds during ultrasound exposure, leading to a decrease in the activity of the extracts. Antimicrobial activity data indicate a greater sensitivity to gram-positive bacteria than to gram-negative bacteria. At the same time, the activity against to phytopathogenic fungi is also more pronounced than against gram-negative bacteria *Erwinia carotovora* spp.

In the case of the gram-positive bacteria *Clavibacter michiganensis*, the highest activity was shown by extracts of JB1A-*m* and JB1A-*usm* (MIC and MBC, 39 µg/mL), as well as extract of JB1E-*m* (MIC, 39 µg/mL; MBC, 78 µg/mL), which leads to the conclusion that the extracts are highly bioactive against this bacterium.

In the case of the gram-negative bacteria *Erwinia carotovora* spp. the highest activity was shown by extracts of JB1A-*m* and JB1A-*usm* (MIC,39 µg/mL; MBC, 39–78 µg/mL), which are classified as very bioactive extracts in relation to this gram-negative bacteria [28]. The remaining extractants of JB1 showed an MIC ranging from 156 to 1250 µg/mL with significant and moderate activity.

For the phytopathogenic fungus *Rhizoctonia solani*, the highest activity was shown by extracts of JB1A-*m* and JB1A-*usm* (MIC, 39–156 µg/mL; MFC, 78–312 µg/mL), and extract of JB1E-*usm* (MIC, 156 µg/mL; MFC, 312 µg/mL).

For the phytopathogenic fungus *Alternaria solani*, the extract of JB1A-*usm* (MIC, 39 µg/mL; MBC, 78 µg/mL) as well as the extract of JB1E-*usm* (MIC, 78 µg/mL; MBC, 156 µg/mL) showed the highest activity. In the case of two phytopathogenic strains, the fungus *Rhizoctonia solani* showed a higher susceptibility to the extracts.

Industrial α-pinene (>98%), tested similarly tested on four strains of microorganisms showed the following values: MIC, 156–625 µg/mL; MBC, 156–1250 µg/mL; and MFC, 312–625 µg/mL. Thus, for all strains, α-pinene showed the greatest activity against the gram-positive bacteria *Clavibacter michiganensis*, and at the same time, this activity was four times less relative to first year berries when using acetone and ethanol during ultrasound-assisted maceration, and similarly when using acetone during maceration.

In the case of the gram-negative bacteria *Erwinia carotovora*, the activity of α-pinene was 16 times lower compared to extracts of JB1A-*m* and JB1A-*usm*. In relation to *Rhizoctonia solani* and *Alternaria solani*, the fungicidal activity of α-pinene was eight times lower compared to the extract of JB1A-*usm*.

Chloramphenicol, which was tested on two phytopathogenic bacterial strains of microorganisms, showed that in the case of the gram-positive bacterium *Clavibacter michiganensis*, its MIC and MBC were more than 2.4 times lower relative to extracts of JB1A-*m* and JB1A-*usm*, as well as to the extract of JB1E-*m*. In the case of the gram-negative bacteria *Erwinia carotovora*, the activity was more than 2.4 times lower relative to extracts of JB1A-*m* and JB1A-*usm.*

### 2.6. Nematocidal Activity

The addition of extracts of immature and mature juniper berries (*J. communis*) to the nematode incubation medium caused a dose-dependent impairment of nematode swimming induced by a mechanical stimulus, for which terms such as the “paralysis” of nematodes is typically used (Table 8). Propylene glycol at a concentrations of 0.025% and 0.05% had no negative effect on the organisms *C. elegans* and *C. briggsae* (Table 8).

Incubation with juniper extracts for three hours did not significantly affect the survival of nematodes of either species. Juniper extracts at a concentration of 0.025% had no negative effect on *C. elegans* organisms. The 70% ethanol and acetone extracts of JB1 at a concentration of 0.025% had no negative effect on *C. elegans* organisms after 24 h of incubation, while a concentration of 0.05% caused disturbances in the swimming motor program and the death of 91.5% and 96.5% of *C. elegans*, respectively. The 70% ethanol extract of JB1 at concentrations of 0.025 and 0.05% caused the death of 85 and 87% of *C. briggsae*, respectively, after 24 h of incubation. As shown in the Table 8, the behavior sensitivity to the actions of extracts for *C. elegans* is significantly higher than *C. briggsae*, in particular for the JB1A extract.

At the same time, at a concentration of 0.05%, the extracts of JB1 did not significantly affect the survival of nematodes of either species during the experiment.

#### The Effect of Berry Extracts on the Resistance of *C. elegans* to Oxidative Stress

Paraquat is characterized by inducing oxidative and toxic stress and is therefore widely used in studies of oxidative stress mechanisms. The preincubation of *C. elegans* with juniper extracts caused the death of 17.3–26.0% of nematodes after 24 h of incubation on NGM (Table 9). In the control variant, the death of nematodes was 2.0%. The treatment of *C. elegans* with 0.5 mM of paraquat caused the death of 20.0% of the nematodes in the control variant.

Paraquat had no significant effect on nematode survival after their treatment with acetone extracts of JB1 and JB2. Treatment with the 70% ethanol extract of JB1 and JB2 increased the proportion of dead nematodes from 17.3 to 69.7% and from 18.7 to 26.7%, respectively. The increase in nematode mortality after co-treatment with paraquat and the 70% ethanol extracts of JB1 and JB2 suggests that these extracts contain substances with pro-oxidant activity, thereby enhancing the toxic effect of paraquat on *C. elegans*.

## 3. Discussion

The present study was developed taking into account the following main aspects: the characteristics of extracts of juniper berries (*J. communis*) of different polarity and extraction conditions and their evaluation as well as the biological profile for potential applications (antimicrobial and antioxidant activity). The evaluation of the extraction efficiency and phytochemical analyses provided the initial idea of the composition of the extract. Changing the extraction method and extractant led to both an increase and a decrease in the extraction yield values [29]. Scientific studies (Damjanovic et al., Proença da Cunha et al., Miceli et al.) found that extraction yields using 70% ethanol, methanol and hexane ranged from 4.4 to 7.6% [30,31,32].

The concentration level of biologically active compounds in juniper berries, which according to the data in the literature have a wide therapeutic potential (phenolics, flavonoids, terpenoids and vitamin C), depends on the genotype, plant part, origin, age, gender, solvent and research conditions [3,33].

Statistical analysis (Post-hoc Tukey Test and Multiple analysis of variance (MANOVA)) identified a significant effect of maturity year, extraction method and solvent on the extract yield and the main phytochemicals (total sugars, total phenolic compounds, total flavonoids, total terpenoids and ascorbic acid content).

Our work on the content of biologically active components of JB1 extracts and in the study of their group composition showed a greater amount of phenolics, flavonoids, terpenoids and ascorbic acid by using 70% ethanol, acetone and methanol during maceration and US compared to extracts of JB2, whose sugar content was the highest when using 70% ethanol. The use of ultrasound had a positive effect on the extraction of phenolics and flavonoids from extracts of JB2, in contrast to extracts of JB1, from which ballast compounds were presumably extracted, as shown by the extraction yield data (Table 1).

The total content of phenolic compounds for JB1 extracts ranged from 1.12 to 56.15 mg equivalent of gallic acid (GAE) per g dry extract; for JB2 extracts, this ranged from 0.93 to 11.1 mg equivalent of gallic acid (GAE) per g dry extract. These data were lower than in the other described works; the total phenolic content for JB2 extracts of *J. communis* from different origins, such as Turkey, Slovakia, Canada, Australia, Portugal, and Romania, varied in the range of 0.19–99 mg equivalent of gallic acid (GAE) per g dry extract (dw) [10,25,31,34,35,36]. Orhan et al. reported that the total phenolic content of JB1 was 130.92 mg equivalent of gallic acid (GAE) per g dry extract [37].

The total flavonoid content for JB1 extracts ranged from 0.89 to 52.31 mg equivalent of Que per g dry extract; for JB2 extracts, this ranged from 0.75 to 9.37 mg Que/g DE. Another work [37] reported a lower flavonoid content of 17.57 mg and 2.56 mg/g dry weight for JB1 and JB2 extracts, respectively.

Total terpenoids ranged from 23 to 133 mg/g DE for JB1 extracts and from 17 to 110 mg/g DE for JB2 extracts. Fejér et al. reported that the total terpenoids in 70% ethanol extract ranged from 38 to 80 mg/g DE, which is comparable to our research data [25].

The content of ascorbic acid for JB1 extracts ranged from 7.1 to 21.3 mg/g DE; for JB2 extracts, this ranged from 1.5 to 2.5 mg/g DE. In other works, its content is 2.1–19 mg/g of dry extract [38,39,40].

The sugar content of JB1 extracts ranged from 4.79 to 21.4 mg/g DE, and that of JB2 extracts, from 10.27 to 53.01 mg/g DE. Nalan et al. reported that the average sugar content of the berries was 25 mg/g DE [41]. Akinci et al. found a sugar content of 21.29 ± 1.47 mg/g DE for JB2 extracts [42].

Currently, a large amount of data has been accumulated on the chemical composition of essential oils of juniper berries (*J. communis*) [3,43], while the composition of extracts has received little attention in the available literature [30,32,44]. The identification of the qualitative and quantitative chemical composition by GC-MS method with different ionization methods of the detectable compounds serves as a reliable and accurate way of interpreting the results obtained and predicting the biological activity of extracts [45,46].

The results of phytochemical analyses of acetone and 70% ethanol extracts of JB1 and JB2 of *J. communis* indicated a high content of bioactive components, including volatile organic compounds (VOCs) in their composition, which could be an indicator of a very good antioxidant, antimicrobial and nematocidal potential [47]. These compounds were mostly secondary metabolites produced by plants to ensure their normal cellular metabolism and provide protection against biotic and abiotic factors and subsequent oxidative damage [48].

The component composition of JB1 extracts using ethanol and acetone in this research work included (in decreasing order) monoterpenes, sesquiterpenes, polysaccharides, steroids, fatty acid esters, bicyclic monoterpenes, monosaccharides, cyclic monoterpenes and acyclic monoterpenoids. For the JB2 extracts, their composition included (in descending order) monosaccharides, monoterpenes, polysaccharides, sesquiterpenes, polyatomic alcohols, α-hydroxyketones, diterpenoids, monoterpene alcohols and monoterpene ketones.

The phytochemical composition results presented here are comparable with the data found in the literature on these extracts. For comparison, Finimundy et al. [49] found that the analysis of the ethanolic extract of JB2 showed that the composition was dominated by monoterpenes (α-pinene: 39.12%; sabinene: 8.87%; β-pinene: 12.68%; myrcene: 12.92%; and limonene: 2.23%) and sesquiterpenes (β-caryophyllene: 4.41%; α-humulene: 1.05%; germacrene D: 4.23%; and δ-cadinene: 1.35%). Höferl et al. [50] found that the monoterpene hydrocarbons α-pinene (51.4%), myrcene (8.3%), sabinene (5.8%), limonene (5.1%) and β-pinene (5.0%) were the main constituents of berries. In other works, monoterpenes, such as α-pinene, β-pinene and β-myrcene, are the most abundant in *J. communis* berries, followed by some sesquiterpene compounds, namely germacrene D [51].

As the berries ripened, a large proportion of secondary metabolites (monoterpenes and sesquiterpenes) decreased, while the concentration of primary metabolites (monosaccharides) increased, which was consistent with other data [51,52]. These works showed that the amount of α-pinene, sabinene, β-pinene and bornyl acetate decreased during berry ripening.

The more ARP, the more effective the antioxidant. The work by Lissi et al. [53] describes two approaches to measuring the total antioxidant capacity while using the CL method, taking into account this feature of the curves—the TRAP method (total reactive antioxidant potential), based the measurement of the latent period of chemiluminescence, and the TAR method (total antioxidant reactivity), based the determination of the magnitude of the quenching of the chemiluminescence intensity.

The analysis of the DPPH radical assay results showed that the tested extracts of JB1, particularly ethanol, acetone and methanol extracts, exhibited the highest antioxidant activity. Depending on the extraction method, the antioxidant activity decreased by 1.5–1.9 times with the use of ultrasound, which may be due to cavitation processes during the use of ultrasound and the intermolecular interaction reactions that occur at the same time. This phenomenon on the extracted material occurred throughout the volume, including inside the cells, increasing the yield of extractive compounds, including those with low antioxidant activity [54,55]. The extracts of JB1 (70% ethanol, acetone and methanol) contained a greater amount of phenolic antioxidants (especially flavonoids and phenolic acids), ascorbic acid, monoterpenes and sesquiterpenes, which is part of the reason for the increase in antioxidant activity.

At the same time, the high content of α-pinene in JB1, as determined by GC-MS, was not the main factor in the antioxidant activity, but only an auxiliary factor. In another work (Höferl et al.), it was reported that the antioxidant activity for individual α- and β-pinene compounds was much lower than for other terpene hydrocarbons, namely sabinene, limonene and myrcene [56]. As for the antioxidant activity of the terpene α-pinene, Berits and his colleagues [57] reported that α-pinene has a significant ability to block lipid peroxidation (IC_50_ value is 0.51 µL/mL).

We suggest that the higher antioxidant activity of JB1E extract compared to JB1A and JB1M extracts may be due to the better solubility of antioxidants in 70% ethanol, which plays an essential role in neutralizing free radicals, thereby increasing its antioxidant capacity, as well as their possible synergism, since the therapeutic effect of phytopreparations is often based on the synergistic effect of their mass components, and the proportion of components in phytopreparations may influence the enhancement of the effect [58].

Studies have shown that gram-positive bacterial strains tend to be more sensitive than gram-negative strains. Brodowska et al. [59] found that the methanolic extracts of *Juniperus communis* berries showed high activity against gram-positive bacteria *S. aureus, E. hirae*, *B. subtilis* (MIC 156.25 µg/mL); the use of the extract on gram-negative strains showed a decrease in activity of about 50% (e.g. *P. aeruginosa* and *Acinetobacter baumannii*) [59,60], while the activity was high when used on fungi (in the case of *Aspergillus niger* (ATCC 15475) and *Penicillium hirsutum*) [60]. The use of methanol and ethyl acetate extracts of *Juniperus communis* from JB2 showed a moderate activity in suppressing the growth of the gram-negative bacteria *P. mirabilis* and *P. vulgaris* as well as gram-positive *S. aureus*, each with an MIC value < 500 µg/mL [61].

The results for the phytopathogens tested showed high activity for extracts of JB1, while in most cases, moderate activity was observed for JB2 for all extractants and extraction conditions. Strains were more sensitive to JB1 extracts, while acetone proved to be the best extractant. Changing the extraction conditions to ultrasound led to an increase in activity against phytopathogenic fungi of 4-fold for MIC and 8-fold for MFC. The lowest activity was shown by extracts of JB2, which may be due to a decrease in the amount of biologically active compounds as the berries ripen.

Based on the work performed, we suggest that the increased activity of JB1 extracts when using acetone and 70% ethanol compared to the antimicrobial activity of pure α-pinene is due to the presence of a synergistic effect between the biologically active components of these extracts [62,63].

The high nematocidal activity of JB1E extracts was demonstrated when assessing their effect on the survival of *C. elegans* and *C. briggsae*. JB1E and JB1A extracts increased the mortality of *C. elegans* and *C. briggsae* by more than 2-fold after 24 h of incubation. At the same time, JB1 extracts at a concentration of 0.025% caused an approximately 4-fold increase in *C. briggsae* mortality.

In addition, the behavioral sensitivity to the action of the acetone extract was revealed in *C. elegans* (96.5%), which turned out to be 46% higher than in *C. briggsae* (Table 8). Therefore, the presence of substances toxic to soil nematodes, especially *C. elegans*, in JB1 extracts could be assumed. This was demonstrated by the high sensitivity of *C. elegans* behavior to the negative effects of *J. communis* extracts. We suggest that the increased activity of JB1 extracts compared to JB2 extracts is related to their higher content of phenolic compounds [64,65], including monoterpenes, especially limonene (0.502–0.546%) [66], sesquiterpenes and other biologically active compounds.

We have not found reliable information on other studies of nematocidal activity in relation to JB1 and JB2 extracts of *J. communis* L. [67]. For other species of the genus *Juniperus*—*J. procera*, *J. excelsa*, *J. phoenicea* and *J. virginiana*, ethanolic extracts of various parts of *J. virginiana* plants had the greatest nematocidal activity against *C. elegans* [68,69]. Extracts from other species of the genus *Juniperus* had weak nematocidal activity. A few articles described the toxic effect of *Juniperus* spp. against plant parasitic nematodes. These include the article by Kong et al. (2006), in which the activity of essential oils of *J. communis*, *J. oxycedrus* and *J. virginiana* against the pine wood nematode *Bursaphelenchus hylophilus* was investigated [70].

The JB1E and JB2E extracts increased the sensitivity of *C. elegans* to paraquat, which is known as a pro-oxidant due to its ability to generate reactive oxygen species [71], and this effect was most pronounced in the JB1E extract. Paraquat is used in agriculture as an herbicide, and the mechanism of its toxic effect is the induction of reactive oxygen species [72]. The increased sensitivity of *C. elegans* to paraquat after pre-exposure to the ethanolic extract of juniper fruits suggests that the mechanism of its nematocidal activity is the induction of oxidative stress.

## 4. Materials and Methods

### 4.1. Materials

The berries of the *Juniper communis* (Figure 4) were collected 7 km from the rural settlement of Chernyshevsky in the Vysokogorsky district of Tatarstan (55.953357° N 49.183376° E). This area is characterized by the absence of industrial facilities. The berries of the first and second year (JB1 and JB2) were harvested in the second or third week of September 2022, during the period of full ripening. The branches of the plant with berries were identified by Doctor of Biological Sciences Firdaus Khazieva (All-Russian Research Institute of Medicinal and Aromatic Plants, Moscow, Russia). The berries were collected randomly from ten trees in a quantity of 0.07–0.1 kg per tree. At the time of picking, the trees were 50-55 years old.

The berries were spherical in shape and had a spicy, slightly balsamic odor, which is characteristic of the species. JB1 are light green in color, 3–6 mm in diameter and 0.03–0.09 g in weight. JB2 are dark blue with a matte waxy sheen, 5–7 mm in diameter and 0.07–0.12 g in weight.

The berries were washed with distilled water to remove dust and dirt, dried with a cotton cloth, and stored in a freezer at a temperature of −35 °C for further study.

All chemicals and reagents used were of analytical grade. Folin-Ciocalteu reagent, 2,2-diphenyl-1-picrylhydrazyl (DPPH) and paraquat dichloride were purchased from Sigma-Aldrich (St. Louis, MO, USA). Gallic acid, chlorogenic acid, caffeic acid and quercetin were purchased from PhytoLab GmbH & Co. KG, Germany (Merck KGaA, Darmstadt, Germany).

### 4.2. Extract Preparation by Maceration

A comparative analysis of the biological properties was carried out as a function of the extraction method and the extractant. The first stage of the work was to obtain extracts from frozen juniper berries by maceration. Frozen JB1 was ground to a paste in a laboratory mill with forced cooling of the grinding chamber (KN 195 Knifetec, Labtec Foss, Hellirod, Denmark) and extracted by single maceration in a flat-bottomed conical flask pre-purged with argon. A series of solvents of increasing polarity were used as extractants—pentane, chloroform, acetone, methanol and 70% ethanol with a biomass/solvent ratio of 1:5. The homogenates were mixed using an automatic magnetic stirrer (RCT basic, IKA, Staufen, Germany) at 500 rpm for 1.5 h at a temperature of 45 °C with a constant supply of dry argon to avoid the oxidation of bioactive compounds [73].

The resulting mixtures were centrifuged to precipitate large particles and destroy the suspension at 10,500 rpm for 15 min at a temperature of 3 °C in order to avoid oxidative processes (H3-18KR centrifuge, Hunan Kecheng Instrument Equipment Co, Ltd., Changsha, China). The extracts were dried on a rotary evaporator (LabTex Re 100-Pro, Labtech Company, Moscow, Russia) over a water bath at a temperature of 31–33 °C and a pressure of 13.3 mbar until the solvent was completely removed. The dried extracts were stored in a medical freezer (Pozis MM-180/20/35, POZIS, Zelenodolsk, Russia) at a temperature of −35 °C for further analysis. Similarly, extracts were obtained from JB2.

### 4.3. Extract Preparation by Ultrasound-Assisted Maceration

The second stage of the work was to obtain juniper berry extracts using ultrasound-assisted maceration. The raw materials, which were ground into a paste, were extracted with a biomass to solvent ratio of 1:3.5 using an I-10-0.63 ultrasonic generator with an immersion probe at a frequency of 22.35 kHz and a power of 290 watts. The extraction was carried out at 45 °C for 15 min with constant stirring [74]. The homogenates were then centrifuged and dried as in the first step. To study the biological properties, the dried extracts were re-dissolved (to obtain a 1% solution of the extracts) in a solvent eutectic consisting of 65% 1,3-propylene glycol; 25% ethanol; and 10% water.

### 4.4. Phisico-Chemical Properties

The pH of juniper berry extracts was determined using a pH meter (HANNA HI 2210, HANNA, Quebec, Canada) by dissolving the dry extract in 70% ethanol. Electrical conductivity was measured in the same solution using a conductivity meter (MARK-603/1, VZOR, Nizhny Novgorod, Russia) [75].

The extraction yield of the extracts was determined gravimetrically. For this purpose, 2 mL of the extract was transferred to a pre-weighed stainless steel bowl and placed in an infrared moisture analyzer (Ohaus MB 25, OHAUS CORPORATION, Parsippany, NJ, USA) at 105 °C until the mass stabilized [27].

### 4.5. Quantification of The Main Phytochemicals

#### 4.5.1. Total Phenolic Compounds Content

The total phenolic content was determined by the spectrophotometric method using the Folin–Ciocalteu reagent [76] with some modifications. For analysis, 0.5 mL of an aliquot of the extract was mixed with 0.5 mL of Folin–Ciocalteu reagent. The solution was kept at 25 °C for 5–8 min, then 2 mL of a 7.5% sodium carbonate solution was added. The volume was increased to 8 mL by adding water, and the resulting solution was incubated in the dark for 60 min. The optical density was then measured at 725 nm. Gallic acid was used as the standard for the calibration curve. The total phenolic content of the samples was quantified using a calibration curve constructed with the standard of gallic acid at different concentrations ranging from 0 to 200 µg/mL, and expressed as mg equivalents of gallic acid per gram of dry extract (mg GAE/g DE).

#### 4.5.2. The Total Content of Flavonoids

The total flavonoid content was determined according to the Stankovich method [77] with some modifications. The bulk of the dry extract was dissolved in 70% ethanol. The aliquot (0.5 mL) was placed in different test tubes, to which 10% aluminum chloride (0.1 mL), 1 M potassium acetate (0.1 mL), 80% methanol (1.5 mL) and distilled water were added. The resulting solutions were mixed and incubated at 25 °C for 30 min. The comparison solution was prepared in the same way, except that a quercetin solution of known concentration was used instead of the extract. The optical density was determined at λ max = 440 nm using a scanning spectrophotometer in the stationary wavelength mode (UV/VIS spectrometer T7DS, Purkinje General Instrument Co., Ltd., Beijing, China) in a cuvette with an optical path length of 10 mm. The concentration of flavonoids was expressed in milligrams of quercetin equivalents per gram of dry extract (mg Que/g DE).

#### 4.5.3. The Total Concentration of Terpenoids

The total concentration of terpenoids was determined directly in freshly prepared extracts by spectrophotometry at a wavelength of 538 nm [78]. For analysis, the extracts were evaporated to remove solvents, followed by their re-extraction with 95% methanol for 5 min at a frequency of 30 Hz at a dry extract/methanol ratio of 1:20. The homogenate was then centrifuged at 10,000 rpm. 1.5 mL of chloroform and 100 mL of concentrated sulfuric acid were added to 200 mL of the filler liquid while cooling. The mixture was incubated for 2 h in the dark, and the upper layer was decanted. The lower red-brown layer was dissolved with 1.5 mL of methanol. The total terpenoid content is calculated relative to the linalool solution with different concentrations ranging from 0 to 200 µg/mL according to the calibration curve and is expressed in mg/g of dry extract (mg/g DE).

#### 4.5.4. The Ascorbic acid Content

The ascorbic acid content in the extract was determined by spectrophotometric method using potassium permanganate as a chromogenic reagent at a wavelength of 530 nm [79]. Standard solutions of ascorbic acid were prepared within the sample range (1 to 100 mg/L), which were dissolved (0.01 g) in a small amount of 0.5% oxalic acid solution and making up with distilled water to obtain a concentration of 100 mg/L. The solution of the chromogenic agent with a concentration of 100 mg/L was prepared by previously dissolving 0.01 g of KMnO_4_ in a 5 molar solution of sulfuric acid in a 100 mL volumetric flask and diluting to the mark with distilled water. 1 mL of chromogenic agent was added to a series of 10 mL standard solutions containing different concentrations of ascorbic acid, and after 5 min, the absorbance of each solution was measured at 530 nm against the blank. The determination of ascorbic acid concentration in dry extracts was carried out in a similar way by replacing the suspension of the standard by an aliquot of the sample dissolved in water.

#### 4.5.5. The Sugar Contents

Total soluble sugars were determined using a spectrophotometer with 3,4-dimethylphenol as the reagent. The initial solutions of sugars—glucose, fructose, lactose and sucrose—were obtained by dissolving 100 mg of sugar in 25 mL of water in 100 mL graduated flasks and diluting to the mark with 0.25% benzoic acid. Working standards (15 mg/mL) were prepared by dilution. A 0.2% solution of 3,4-dimethylphenol was dissolved in 50% ethanol. The aliquot of the sample containing 15–150 µg of sugar was placed in a 25 mL calibrated and graduated tube. The solution was evaporated in a water bath to 1 mL and then mixed with 1 mL of the 3,4-dimethylphenol solution. Then, 72% sulfuric acid was added drop by drop. The absorption of the glucose, fructose and lactose solutions was detected at 510 nm and that of sucrose at 520 nm in a cuvette with an optical path length of 10 mm for 1 measurement in a scanning mode with a limit of 460–570 nm. The solution for the determination of sugars in the extract was obtained by dissolution in 50% ethanol. Total soluble sugars were calculated as the sum of reducing and non-reducing sugars and was expressed as mg/g of dry extract [80].

#### 4.5.6. GS-MS Analysis

The qualitative and quantitative composition of JB1 and JB2 extracts of the common juniper *J. communis* was determined by gas chromatography–mass spectrometry (GC-MS) on a Crystal 5000.2 gas chromatograph using a quadrupole mass spectrometer (Chromatek, Yoshkar-Ola, Russia) with electron ionization (EI, 70 eV) in the range of *m*/*z* 50–550.

For extraction, berries weighing 10 g were selected, lyophilized at −80 °C for 20 h (BK-FD12P, Biobase, Jinan, China), and ground to a particle size of approximately 0.1–0.5 mm using a laboratory mill with forced cooling of the grinding chamber (KN 195 Knifetec, Labtec Foss, Hilleroed, Denmark). The extraction was then performed by maceration by dissolving the powder (1 g) in 10 mL of 99.5% ethanol or 99.9% acetone and mixing at 500 rpm using an automatic magnetic stirrer (Ohaus Guardian 7000, OHAUS CORPORATION, Parsippany-Troy Hills, New Jersey, USA) at a temperature of 45 °C for 1.5 h with a constant supply of dry argon to avoid the oxidation of bioactive compounds [68]. The resulting homogenate was centrifuged to precipitate large particles and suspensions at 11,000 rpm for 15 min at a temperature of +5 °C in order to avoid oxidative processes (H3-18KR centrifuge, Hunan Kecheng Instrument Equipment Co, Ltd., China). They were then filtered through Whatman filter paper Grade 41 together with sodium sulphate to remove sediment and traces of water on the filter paper [81]. Before filtering, the filter paper was moistened with absolute alcohol or acetone together with sodium sulphate. The filtrate was then concentrated on a rotary evaporator (LabTex Re 100-Pro, Labtech Company, Moscow, Russia) in a water bath at a temperature of 30–32 °C and a pressure of 13.3 mbar to a concentration of 1 with dry argon passed through the solution. For further purification, the extract was passed through a 0.45 micron CHROMAFIL Xtra filter (Macherey-Nagel, Düren, Germany). The extracts contained both polar and non-polar components of the plant material, and 1 µL of the solution sample was used in GC-MS for the analysis of various compounds [82].

The GC-MS analysis of extracts of immature and mature juniper berries was carried out without additional dilution. The chromatographic separation conditions were chosen on the basis of known data [83], which indicate that the composition of the extract of most plants contains components that differ in thermal stability, chromatographic lability, polarity and boiling point over a wide range.

Chromatographic separation was performed using a TG-5MS quartz capillary column (5%-phenyl-, 95%-dimethyl polysiloxane, 30 m × 0.25 mm × 0.25 microns, Thermo Fisher Scientific, Waltham, MA, USA). GC-MS analysis conditions: injector temperature, 280 °C; interface temperature, 270 °C; ion source temperature, 250 °C; initial column thermostat temperature, 75 °C; initial temperature control time, 2 min; column temperature increased at a rate of 10 °C/min; final column temperature, 280 °C; and volume velocity of the gas carrier (He, 99.99%), 0.9 mL/min at a constant flow. The sample volume is 1 µL, and the flow split is 1:100. Mass spectral data were processed using Chromatek Analyst software (Chromatek, Yoshkar-Ola, Russia), NIST MS Search Program, V.2.3 (NIST, Gaithersburg, MD, USA), and electronic mass spectral libraries NIST’20 (NIST, Gaithersburg, MD, USA) and Wiley (12th edition, Wiley, USA).

### 4.6. Biological Activity Analysis

#### 4.6.1. DPPH-Free Radical Scavenging Assay

The assessment of the antioxidant activity is very specific to the sample and method since the reaction mechanisms of the methods for determining antioxidant activity are completely different in terms of the types of oxidants, reaction conditions and even results.

The following two categories of methods have been used to determine the antioxidant activity of juniper berries: (a) the spectrophotometric method, based on the reactions of the colored reagent 2,2-diphenyl-1-picrylhydrazyl radical (DPPH), and (b) the chemiluminescence method (CL), based on the encapsulating or inhibiting effect of antioxidants on the luminol-enhanced chemiluminescence (CL) [84,85].

The chemiluminescent assay is described in [26] and adapted to the Lum-1200 luminometer (LLC DISoft, Moscow, Russia) [86]. The results were processed on a personal computer using the PowerGraph and OriginLab software. Luminol solution (Alfa Aesar, Ward Hill, MA, USA) (1 mmol L^−1^) was prepared by dissolving luminol in aqueous 0.1 M NaOH. Immediately prior to analysis, the luminol stock solution was diluted 4-fold with distilled water.

Chemiluminescence analysis was performed as follows: thermostated at 30 °C; and cell was charged with 1 mL of the reaction mixture containing 400 μL of 250 μM luminol solution, 500 μL of 0.5 M of Tris buffer (Fisher Chemical, Loughborough, UK), pH 8.6, and 100 μL of 40 мM aqueous 2,2’-azobis(2-methylpropionamidine) dihydrochloride (AAPH) solution (Acros Organics, Saint Louis, MO, USA). The basic chemiluminescence signal was measured for 10 min, then 10 μL of the solution of the test compound was added, after which the chemiluminescence signal was acquired.

The analysis of DPPH (2.2-diphenyl-1-picrylhydrazyl) was carried out according to the method of Brand-Williams et al. [71] with some modifications. The initial solution was prepared by dissolving 24 mg of DPPH in 100 mL of ethanol and then stored at −20 °C. The working solution was obtained by mixing 10 mL of the initial solution with 45 mL of ethanol to give an optical density of 1.1 ± 0.02 units at 515 nm using a spectrophotometer. Berry extracts (200 µL) were incubated with 800 µL of DPPH solution for 30 min in the dark. The optical density was then measured at 517 nm. The extracts activeness to remove the chromogenic radical was expressed by the EC_50_ value (mg/mL), i.e., the concentration required to reduce the concentration of DPPH by 50%. The effectiveness of the antioxidant was defined as the antiradical power (ARP), defined as 1/EC_50_.

#### 4.6.2. Antimicrobial Activity

The following test strains of microorganisms were used in this work: phytopathogenic bacteria—*Clavibacter michiganensis* gram-positive strain VKM Ac-1404 (VKM IBFM RAS, Pushchino, Russia) and gram-negative bacterial strain *Erwinia carotovora* spp. SCC3193, a phytopathogenic fungi—*Alternaria solani* K-100054 (VNIIF, Bolshiye Vyazemy, Russia) and *Rhizoctonia solani* BKM F-895 (VKM IBFM RAS, Pushchino, Russia).

Chloramphenicol for phytopathogenic bacterial strains (Kazan Pharmaceutical Plant, Kazan, Russia) and difenoconazole for fungal strains (Score 250 EC, Syngenta, Basel, Switzerland) were used as reference compounds in the experiments.

Microorganisms were cultured in standard sterile nutrient broths—BTN broth, Hottinger medium and potato dextrose agar. Bacterial concentrations were determined according to standard protocols using a DEN-1B densitometer (Biosan, Riga, Latvia). Microorganisms were incubated in a thermostat for 5 days at 30 °C for *Clavibacter michiganensis* VKM Ac-1404 and 28 °C for *Erwinia carotovora* spp. SCC3193, *Alternaria solani* K-100054 and *Rhizoctonia solani* VKM F-895., respectively. All analyses were performed in triplicate.

The results are presented in the table in terms of minimum inhibitory concentrations (MIC)—concentrations that stop the growth of bacteria and fungi, and minimum bactericidal and fungicidal concentrations (MBC and MFC, respectively)—i.e., concentrations that cause cell death. The experiments were designed to find antimicrobial activity, which was determined by the serial dilution method according to the methods described in [87,88]. In the experiments, the minimum inhibitory concentration (MIC) was determined by the double serial dilution method. The fungistatic activity of the alcoholic extract was measured by serial dilution in liquid medium. The extracts were tested at concentrations from 4.88 to 2500 μg/mL. A bacterial suspension or a piece of fungal mycelium was added to each tube containing an extract of known concentration. After incubation, microbial viability was assessed visually, and the minimum concentration that contributed to the cessation of growth of the culture without killing it (minimum inhibitory concentration) was determined. To determine the minimum bactericidal and fungicidal concentrations of the extract, bacterial inoculum or pieces of fungal mycelium taken from all tubes without visible growth were added to Petri dishes containing agarified nutrient medium using a bacteriological loop. The minimum concentration at which bacteria were killed was considered the minimum bactericidal concentration, and for fungi the minimum fungicidal concentration.

The inoculum was prepared from a daily culture of microorganisms grown in a liquid nutrient medium. Uniform, isolated colonies grown on cut agar were selected. A small amount of material was transferred with a loop from the tops of the colonies into a test tube containing sterile saline solution (concentration—0.09%), and the inoculum density was adjusted to 0.5 units according to the McFarland standard. The optical density of the suspension was monitored using a densitometer “DEN-1B” (Biosan, Riga, Latvia) that was calibrated by the manufacturer within the range of 0.0–6.0 McFarland units. The number of bacteria in the inoculum was 1.5 × 10^8^ CFU/mL. Test tubes containing only culture media were used as controls.

The minimum inhibitory concentrations were determined in test tubes. To determine the MIC, the test substance was added to the first tube containing 2 mL of liquid nutrient medium, then 1 mL of the solution was removed from the first tube and transferred to a second tube containing 1 mL of the medium. This procedure was carried out in the range of 10 successive concentrations in order to obtain a series of dilutions of the substances under investigation. Further, a bacterial suspension in the amount of 30 µL was also added to each tube.

In addition, to determine the minimum bactericidal concentrations of the substances studied, 10 µL of the inoculum taken from the test tubes with no visible growth from the MIC determination experiment were streaked onto Petri dishes with an agar nutrient medium using a bacteriological loop.

As an additional comparison drug, a natural antimicrobial agent, α-pinene, which was the major constituent of the berry extracts according to GC/MS data analysis, was used as a positive control. All samples were tested for the effects of phytopathogenic bacterial strains causing ring rot (*Clavibacter michiganensis*), the bacterial cancer of nightshade and beetroot (*Erwinia carotovora* spp.), and phytopathogenic-fungi-causing rhizoctoniosis (*Rhizoctonia solani* VKM F-895) and alternariosis (*Alternaria solani* K-100054) [45,89].

According to the classification presented in [28], plant extracts with MIC values of less than 100 µg/mL should be considered as very active; in the range of 100–512 µg/mL, as having significant activity; in the range of 512–2048 µg/mL, as having moderate activity; more than 2048 µg/mL, as being inactive. It is suggested in [90] that the extracts of medicinal plants with MIC values < 160 µg/mL should be considered as promising when considering antimicrobial activity.

#### 4.6.3. Nematocidal Activity

The *Caenorhabditis elegans* wild-type strain N2 and *Caenorhabditis briggsae* wild-type strain AF16, obtained from the Caenorhabditis Genetics Centre (CGC) (University of Minnesota, Minneapolis, MN, USA), were used in this work. Nematodes were grown in Petri dishes (100 mm diameter) with a standard growth medium (NGM) (3 g/LNaCl; 17 g/L bacto agar; 2.5 g/L bactopepton; 5 mg/L cholesterol; 1 mM CaCl_2_; 1 mM MgSO_4_; and 25 mM potassium phosphate buffer (pH 6.0)). *Escherichia coli* OP50, obtained from CGC, was used to feed nematodes [91]. Experiments were performed at 22 °C with young adults (within the first day after the L4 larval stage).

For each experiment, nematodes were washed from the agar surface with an M9 buffer (3 g/L KH_2_PO_4_, 6 g/L Na_2_HPO_4_, 5 g/L NaCl and 1 mM MgSO_4_) into a 40 mm diameter Petri dish and pipetted into a 10 mL glass centrifuge tube. In this tube, the nematodes were washed to remove the growth medium, bacteria and exometabolites. This was achieved by adding 10 mL of M9 buffer to the tube. After the nematodes had settled to the bottom of the tube, the supernatant was removed. This process was repeated three times. The total washing time was approximately 30 min [92].

To assess the toxicity of the juniper extracts, nematodes were transferred to clean 10 mL glass centrifuge tubes (50 worms in each tube). After the worms had settled to the bottom of the tubes, the supernatant was removed, and M9 buffer and juniper extracts were added. The final concentrations of juniper extracts were 0.025 and 0.05% and the final volume of incubation medium was 1 mL. Propylene glycol at concentrations of 0.025 and 0.05% was used as a negative control. The death of nematodes within 24 h was used as a criterion for the toxic effect of extracts of immature and mature *J. communis* berries. Nematodes lacking both spontaneous locomotion and motor activity in response to a mechanical stimulus were considered dead. The death of *C. elegans* and *C. briggsae* was recorded using an SMZ-05 stereomicroscope. All experiments were performed in quadruplicate, with 50 nematodes per extract. The total number of nematodes in each replicate was 200.

To determine the resistance of *C. elegans* to oxidative stress, nematodes were washed to remove growth medium, bacteria and exometabolites, as described above, and 50 specimens were placed in clean 10 mL glass centrifuge tubes. M9 buffer and the extracts under investigation were added to these tubes at a concentration of 0.05%. The final volume of the incubation medium was 1 mL. Juniper extracts were not added to the control tubes. The nematodes were incubated for 1 h at a temperature of 22 °C. The nematodes were then washed twice with 10 mL of M9 buffer. After washing, M9 buffer and the herbicide paraquat (paraquat dichloride, Sigma-Aldrich, Burlington, Massachusetts, USA) were added to the test tubes [93]. The final volume of the incubation medium was 1 mL, and the concentration of paraquat was 0.5 mM. Nematodes were incubated with paraquat for 1 h at 22 °C, washed twice again with 10 mL of M9 buffer, and transferred to 40 mm diameter Petri dishes containing nematode growth medium and *E. coli* OP50. After 24 h of incubation at 22 °C, the number of dead nematodes was calculated. All experiments were performed in triplicate, with 50 nematodes for each extract. The total number of nematodes in each replicate was 150. 

The percentage of paralyzed nematodes was calculated for each juniper extract. Table 8 shows the mean percentage for 4 experiments, and Table 9 shows the mean percentage for 3 experiments with standard error.

### 4.7. Statistical Analysis

The results were summarized as the mean values ± standard deviation (SD). The data were processed using Microsoft Excel 2016 and OriginPro 9.5 software (OriginLab Co., Northampton, MA, USA). The phytochemical data of the JB1 and JB2 extracts were compared pairwise using Student’s *t*-test (normal distribution, Kolmogorov–Smirnov and Shapiro–Wilk criteria, *p* > 0.05). Repeated measures using analysis of covariance (ANOVA) and Post-hoc Tukey test were used to examine the extraction yield and the main phytochemicals (total sugars, total phenolic compounds, total flavonoids, total terpenoids and ascorbic acid content) between berries of the 1st and 2nd year of maturity. Data analysis was performed using IBM SPSS 23.0 statistical software (Chicago, IL, USA). The level of *p* < 0.05 was considered statistically significant. Pearson correlations (*p* < 0.05) were used to identify correlations between phytochemicals and antioxidant activity of acetone, methanol and 70% ethanol extracts of JB1 and JB2.

The statistical processing of nematocidal activity results was performed using Fisher’s angular transformation. The regression analysis of calibration characteristics and the mathematical prediction of the peak area response for chromatographic analysis were performed using OriginPro 9.5 software (Origin Lab Corp, Northampton, MA, USA) and processed according to standards [94].

## 5. Conclusions

The presented research shows the results of the group composition and bioactivity of the extracts of juniper berries (*J. communis*) of 1- and 2- years of maturity collected in the temperate continental zone of the Republic of Tatarstan, the Russian Federation, using different solvents (pentane, chloroform, acetone, methanol and 70% ethanol) and extraction methods—maceration or ultrasound-assisted maceration. The group composition of JB1 and JB2 extracts, obtained with the help of different extractants and conditions, differed in their quantitative composition. The differences observed in the phytopathogenic activity of the extracts depended on the choice of extractants and maceration conditions. The highest antioxidant and antimicrobial activity corresponded to acetone and 70% ethanol extracts of 1-year-old berries.

Furthermore, the individual chemical composition and nematocidal activity were determined for the 70% ethanol and acetone extracts of berries that previously showed a higher biological profile. A total of 38 individual compounds were identified depending on the extractant used (24 compounds in second-year berries and 23 compounds in first-year berries). According to GC-MS data, the main major compounds were α-pinene (20.6–42.3%), dihydroxyacetone (0.6–39.8%), myo-inositol (4.8–19.1%), germacrene D (4.1–14.8%) and β-myrcene (3.6–8.5%). The nematocidal activity data indicated the presence of substances toxic to soil nematodes, especially for *C. elegans*, in the extracts of 1-year-old berries, and that in the presence of paraquat, the activity increased several times.

According to the results of this study, it is suggested that the 70% ethanol and acetone extracts of the first-year berries of *J. communis* can be alternatives to conventional synthetic pesticides and could be used in the development of biopesticide products after further necessary studies, including further identification of the components responsible for their high phytopathogenic activity.

## Figures and Tables

**Figure 1 plants-12-03401-f001:**
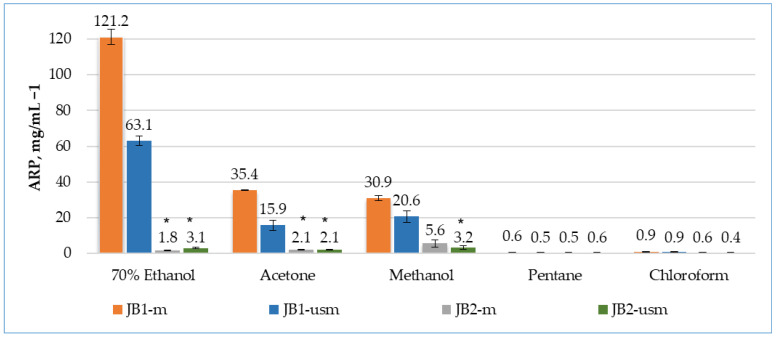
Analysis of the antioxidant activity of berry extracts by maceration and using ultrasound using the DPPH method. The effectiveness of the antioxidant was defined as antiradical power (ARP), defined as 1/EC_50_. * *p* < 0.05, significance of differences compared to JB1 extracts; JB1-*m*: first year, maceration; JB1-*usm*: first year, ultrasound-assisted maceration; JB2-*m*: second year, maceration; JB2-*usm*: second year, ultrasound-assisted maceration.

**Figure 2 plants-12-03401-f002:**
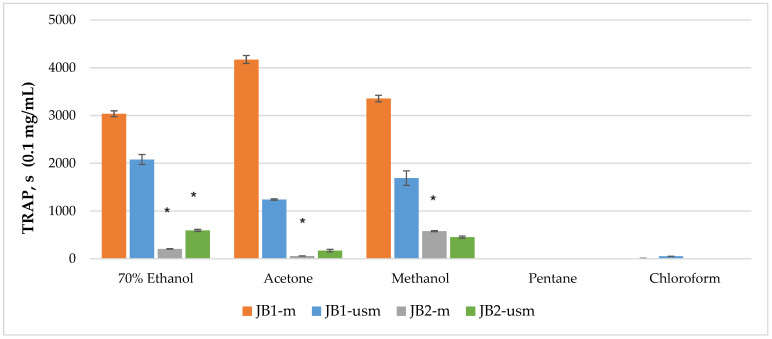
TRAP (total reactive antioxidant potential) index of chemiluminescent analysis of berry extracts by maceration and ultrasound-assisted maceration; * *p* < 0.05, significance of differences compared to JB1 extracts; JB1-*m*: first year, maceration; JB1-*usm:* first year, ultrasound-assisted maceration; JB2-*m*: second year, maceration; JB2-*usm:* second year, ultrasound-assisted maceration.

**Figure 3 plants-12-03401-f003:**
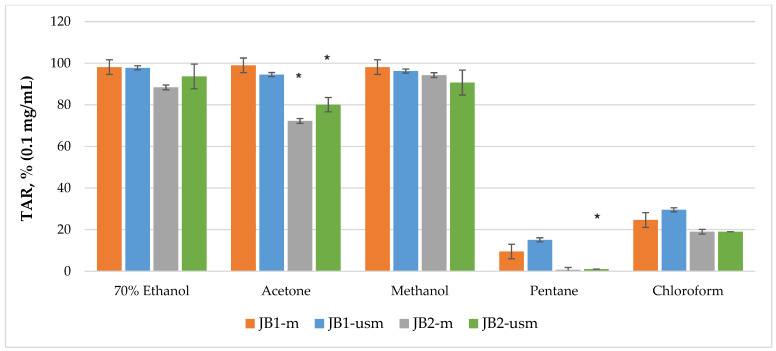
TAR (total antioxidant reactivity) index of chemiluminescent analysis of berry extracts by maceration and ultrasound-assisted maceration; * *p* < 0.05, significance of differences compared to JB1 extracts; JB1-*m*: first year, maceration; JB1-*usm*: first year, ultrasound-assisted maceration; JB2-*m*: second year, maceration; JB2-*usm*: second year, ultrasound-assisted maceration.

**Figure 4 plants-12-03401-f004:**
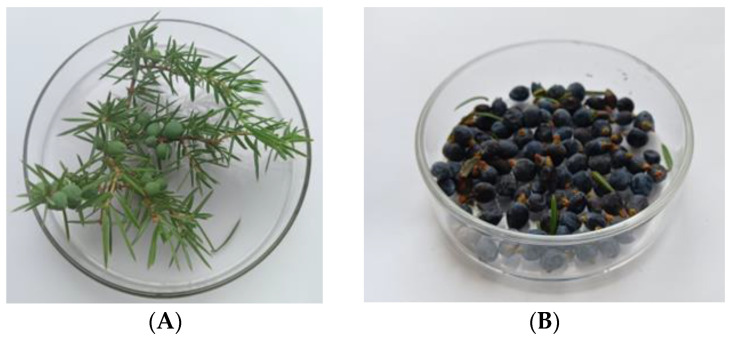
Conifers with JB1 (**A**) and JB2 (**B**).

**Table 1 plants-12-03401-t001:** Extraction yield, pH and electrical conductivity of JB1 and JB2 extracts.

Conditions	Extraction Yield (%) ^3^		pH		Electrical Conductivity, mS/cm.	
	^2^ JB1	JB2	JB1	JB2	JB1	JB2
P-*m* ^1^	** 2.21 ± 0.09	* 0.63 ± 0.07	6.57 ± 0.07	6.71 ± 0.05	11.31 ± 0.02	9.61 ± 0.07
P-*usm*	** 2.91 ± 0.15	** 0.87 ± 0.09	7.15 ±0.09	6.79 ± 0.03	10.17 ± 0.03	7.55 ± 0.05
C-*m*	1.38 ± 0.15	1.21 ± 0.13	6.33 ± 0.02	6.61 ± 0.01	23.67 ± 0.08	17.25 ± 0.03
C-*usm*	* 1.87 ± 0.21	* 1.61 ± 0.11	6.51 ± 0.05	6.75 ± 0.03	17.65 ± 0.07	15.55 ± 0.03
A-*m*	* 3.15 ± 0.11	* 3.23 ± 0.21	6.17 ± 0.03	6.33 ± 0.09	21.13 ± 0.05	15.33 ± 0.09
A-*usm*	4.21 ± 0.17	4.33 ± 0.17	6.21 ± 0.07	6.51 ± 0.05	19.81 ± 0.05	13.97 ± 0.11
M-*m*	* 4.53 ± 0.23	* 3.67 ± 0.18	5.53 ± 0.05	6.18 ± 0.03	43.15 ± 0.05	33.17 ± 0.02
M-*usm*	* 5.74 ± 0.11	* 4.63 ± 0.19	5.31 ± 0.05	6.33 ± 0.07	57.35 ± 0.05	25.63 ± 0.7
E-*m*	* 5.57 ± 0.19	* 6.73 ± 0.31	5.17 ± 0.03	5.76 ± 0.05	39.76 ± 0.03	35.55 ± 0.05
E-*usm*	* 7.33 ± 0.19	* 8.79 ± 0.15	4.97 ± 0.05	5.91 ± 0.03	43.75 ± 0.03	33.51 ± 0.03

^1^ P-*m*—Pentane, maceration; P-*usm*—pentane, ultrasound-assisted maceration; C-*m*—chloroform, maceration; C-*usm*—chloroform, ultrasound-assisted maceration; A-*m*—acetone, maceration; A-*usm*—acetone, ultrasound-assisted maceration; M-*m*—methanol, maceration; M-*usm*—methanol, ultrasound-assisted maceration; E-*m*—70% ethanol, maceration; E-*usm*—70% ethanol, ultrasound-assisted maceration; ^2^ JB1—Juniper berries in the first year of maturation; JB2—Juniper berries in the second year of maturation; ^3^ Each value represents the mean ± SD; * statistical differences at *p* < 0.05 probability level of JB1 relative to JB2; **: statistical differences at *p* < 0.01 probability level of JB1 relative to JB2.

**Table 2 plants-12-03401-t002:** The total content of substances of JB1 and JB2 extracts.

Conditions	Total Sugars, mg/g DE ^3^		Total Phenolic Compounds ^4^, mg GAE/g DE		Total Flavonoids ^5^, mg Que/g DE		Total Terpenoids ^6^, mg/g DE		Content of Ascorbic Acid, mg/g DE	
	^2^ JB1	JB2	JB1	JB2	JB1	JB2	JB1	JB2	JB1	JB2
P-*m* ^1^	nd ^7^	nd	* 1.31 ± 0.15	* 0.93 ± 0.15	* 0.97 ± 0.29	* 0.87 ± 0.27	* 29 ± 1.5	* 17 ± 1.3	nd	nd
P-*usm*	nd	nd	1.12 ± 0.13	1.18 ± 0.19	* 0.89 ± 0.27	* 0.75 ± 0.27	* 23 ± 1.7	* 19 ± 1.2	nd	nd
C-*m*	nd	0.151 ± 0.17	* 5.81 ± 0.33	* 4.23 ± 0.21	* 5.11 ± 0.67	* 3.97 ± 0.42	* 32 ± 1.9	* 19 ± 2.1	nd	nd
C-*usm*	nd	0.217 ± 0.19	* 5.43 ± 0.31	* 3.93 ± 0.17	* 4.71 ± 0.17	* 3.61 ± 0.23	* 28 ± 1.7	* 18 ± 2.3	nd	nd
A-*m*	** 4.79 ± 0.11	** 13.11 ±0.11	* 25.67 ± 0.27	* 8.55 ± 0.19	* 23.57 ± 0.53	* 7.21 ± 0.19	* 116 ± 1.2	* 110 ± 1.9	* 11.7 ± 0.11	* 1.5 ± 0.11
A-*usm*	** 5.36 ± 0.1	** 14.61 ±0.13	* 17.49 ± 0.29	* 8.49 ± 0.25	* 10.57 ± 0.31	* 7.01 ± 0.35	* 123 ± 2.7	* 89 ± 2.1	* 7.1 ± 0.15	* 1.5 ± 0.17
M-*m*	* 5.53 ± 0.21	* 10.27 ± 0.23	* 23.33 ± 0.25	* 11.01 ± 0.23	* 21.52 ± 0.43	* 9.13 ± 0.21	* 126 ± 2.7	* 87 ± 1.2	* 10.1 ± 0.21	* 2.1 ± 0.31
M-*usm*	* 6.33 ± 0.19	* 11.53 ± 0.25	* 19.97 ± 0.21	* 10.05 ± 0.33	* 18.52 ± 0.47	* 8.13 ± 0.51	* 129 ± 3.5	* 91 ± 2.3	* 9.2 ± 0.15	* 2.3 ± 0.19
E-*m*	** 19.27 ± 0.25	** 47.32 ± 0.33	* 56.15 ± 0.29	* 7.71 ± 0.17	** 52.31 ± 0.33	** 6.37 ± 0.37	* 133 ± 2.3	* 57 ± 1.5	* 35.3 ± 0.12	* 2.1 ± 0.21
E-*usm*	* 21.39 ± 0.23	* 53.01 ± 0.31	* 44.31 ± 0.33	* 10.84 ± 0.15	* 40.33 ± 0.33	* 9.37 ± 0.37	* 130 ± 2.3	* 61 ± 1.5	* 21.3 ± 0.19	* 2.5 ± 0.15

^1^ P-*m*—Pentane, maceration; P-*usm*—pentane, ultrasound-assisted maceration; C-*m*—chloroform, maceration; C-*usm*—chloroform, ultrasound-assisted maceration; A-*m*—acetone, maceration; A-*usm*—acetone, ultrasound-assisted maceration; M-*m*—methanol, maceration; M-*usm*—methanol, ultrasound-assisted maceration; E-*m*—70% ethanol, maceration; E-*usm*—70% ethanol, ultrasound-assisted maceration; ^2^ JB1—Juniper berries in the first year of maturation; JB2—Juniper berries in the second year of maturation; ^3^ Each value represents the mean ± SD; ^4^ Total phenolic compounds per gallic acid equivalent (mg GAE/g DE); ^5^ Total flavonoids per quercetin equivalent (mg Que/g DE); ^6^ Total terpenoids per linalool equivalent (mg/g DE); ^7^ nd: not detected, * statistical differences at *p* < 0.05 probability level of JB1 relative to JB2; **: statistical differences at *p* < 0.01 probability level of JB1 relative to JB2.

**Table 3 plants-12-03401-t003:** Significance of the influence of the factors on the extraction yield and the main phytochemicals (total sugars, total phenolic compounds, total flavonoids, total terpenoids and content of ascorbic acid) according to the Post-hoc Tukey test (*p* < 0.05 is considered significant).

Factor	Extraction Yield	Total Sugars	Total Phenolic Compounds	Total Flavonoids	Total Terpenoids	Content of Ascorbic Acid
Extraction method (*m* and *usm*) ^1^	0.000	0.000	0.000	0.000	0.000	0.000
Year of maturation (JB1 and JB2) ^2^	0.000	0.000	0.000	0.000	0.000	0.000
Solvents (P, C, A, M and E) ^3^	0.000	0.000	0.000	0.000	0.000	0.000

^1^ *m*—Maceration; *usm*—ultrasound-assisted maceration; ^2^ JB1—Juniper berries in the first year of maturation; JB2—Juniper berries in the second year of maturation; ^3^ P—pentane; C—chloroform; A—acetone; M—methanol; E—70% ethanol.

**Table 4 plants-12-03401-t004:** GC-MS identification and quantification of the components of JB1 and JB2.

No ^2^	*t_R_ *^3^, (min)	Component	^4^ ω, %				Class of Compounds
			^1^ JB1E *	JB1A *	JB2E *	JB2A *	
1.	3.962	1-Hydroxybut-3-en-2-one	nd ^4^	nd	1.913 ± 0.005	nd	ketone
2.	4.063	Acetol	nd	nd	3.439 ± 0.009	0.828 ± 0.006	α-hydroxy ketones
3.	4.199	2-Hydroxy-3-oxobutanal	nd	nd	2.427 ± 0.007	nd	beta-ketoaldehydes
4.	4.675	Glyceraldehyde	nd	nd	0.785 ± 0.004	3.548 ± 0.009	Monosaccharide
5.	5.288	Methyl 2-oxopropanoate	nd	nd	1.131 ± 0.006	nd	Keto acid
6.	5.507	Dihydroxyacetone	0.625 ± 0.005	1.414 ± 0.003	39.830 ± 0.009	17.487 ± 0.009	Monosaccharide
7.	6.016	2-Cyclopenten-1-one, 2-hydroxy-	nd	nd	0.642 ± 0.003	nd	Lactone
8.	6.310	α-Pinene	42.272 ± 0.011	40.944 ± 0.013	20.558 ± 0.008	40.388 ± 0.014	Monoterpene
9.	6.783	1,2,3-Propanetriol	nd	nd	1.868 ± 0.004	7.337 ± 0.008	Polyalcohol
10.	6.883	Sabinene	0.564 ± 0.008	0.482 ± 0.004	nd	nd	Bicyclic monoterpene
11.	6.944	2-Hydroxy-γ-butyrolactone	nd	nd	1.279 ± 0.011	nd	Lactone
12.	7.033	β-Myrcene	8.467 ± 0.011	7.491 ± 0.009	3.553 ± 0.005	6.411 ± 0.007	Monoterpene
13.	7.456	δ-3-Carene	1.155 ± 0.002	1.063 ± 0.005	nd	nd	Bicyclic monoterpene
14.	7.757	Limonene	0.546 ± 0.006	0.502 ± 0.004	nd	nd	Cyclic monoterpene
15.	8.284	4-Methyl-1H-pyrazole-3-carboxylic acid	nd	nd	0.829 ± 0.007	nd	Pyrazole
16.	9.574	Verbenol	nd	nd	0.964 ± 0.005	0.852 ± 0.006	Monoterpene alcohol
17.	10.341	2H-Pyran-2-methanol, tetrahydro-	nd	nd	0.911 ± 0.008	nd	Pyran
18.	10.517	Verbenone	nd	nd	0.324 ± 0.002	0.865 ± 0.004	Monoterpene ketone
19.	10.782	1,2,3-Propanetriol, 1,2-diacetate	nd	nd	0.772 ± 0.003	nd	Lipid (diacylglycerol)
20.	11.033	Methyl citronellate	0.428 ± 0.006	0.470 ± 0.004	nd	nd	Acyclic monoterpenoid
21.	11.581	Bornyl acetate	nd	0.438 ± 0.002	nd	nd	Bicyclic monoterpenoid
22.	13.061	β-Elemene	0.607 ± 0.003	0.393 ± 0.004	0.534 ± 0.003	nd	Sesquiterpene
23.	13.552	Caryophyllene	2.427 ± 0.011	2.241 ± 0.009	1.053 ± 0.007	1.081 ± 0.008	Sesquiterpene
24.	13.717	(E)-β-Farnesene	0.375 ± 0.003	0.347 ± 0.004	nd	nd	Sesquiterpene
25.	14.007	α-Humulene (α-Caryophyllene)	1.720 ± 0.011	1.537 ± 0.012	0.546 ± 0.009	0.935 ± 0.008	Sesquiterpene
26.	14.312	Germacrene D	14.440 ± 0.010	14.769 ± 0.013	4.132 ± 0.007	6.823 ± 0.003	Sesquiterpene
27.	14.487	Bicyclogermacrene	0.493 ± 0.003	nd	nd	nd	Sesquiterpene
28.	15.444	Germacrene D-4-ol	1.766 ± 0.008	1.941 ± 0.009	0.857 ± 0.007	1.220 ± 0.005	Sesquiterpene
29.	17.153	Myo-inositol	19.172 ± 0.014	12.673 ± 0.009	11.265 ± 0.011	4.766 ± 0.007	Polysaccharide
30.	21.504	Epimetendiol	1.891 ± 0.005	3.654 ± 0.007	nd	nd	Steroid
31.	21.514	1,3,6,10-Cyclotetradecatetraene, 3,7,11-trimethyl-14-(1-methylethyl)-, [S-(E,Z,E,E)]-	nd	nd	nd	1.786 ± 0.004	Cembrane diterpenoids
32.	22.908	Isocembrol	nd	nd	0.388 ± 0.008	1.221 ± 0.006	Diterpenoid
33.	22.923	(1R,4aR,5S)-5-((E)-5-Methoxy-3-methylpent-3-en-1-yl)-1,4a-dimethyl-6-methylenedecahydronaphthalene-1-carbaldehyde	0.754 ± 0.007	1.830 ± 0.008	nd	nd	Aromatic aldehyde
34.	23.159	Dehydroabietinol	nd	0.721 ± 0.004	nd	nd	Diterpenoid
35.	23.783	Cyclopropaneoctanoic acid, 2-[[2-[(2-ethylcyclopropyl)methyl]cyclopropyl]methyl]-, methyl ester	0.948 ± 0.003	2.525 ± 0.008	nd	nd	Fatty acid ester
36.	23.933	(1R,4aR,5S)-5-[(E)-5-Hydroxy-3-Methylpent-3-enyl]-1,4a-dimethyl-6-methylidene-3,4,5,7,8,8a-hexahydro-2H-naphthalene-1-carbaldehyde	0.775 ± 0.005	1.899 ± 0.011	nd	nd	Cyclic aldehyde
37.	25.725	Methyl 5,11,14-eicosatrienoate	0.575 ± 0.004	nd	nd	nd	Fatty acid esters
38.	34.863	Nonacosan-10-ol	nd	2.666 ± 0.012	nd	4.452 ± 0.015	Fatty alcohol

^1^ JB1E—Juniper berries in the first year of maturation, 70% ethanol; JB1A—Juniper berries in the first year of maturation, acetone; JB2E—Juniper berries in the second year of maturation, 70% ethanol; JB2A—Juniper berries in the second year of maturation, acetone; ^2^ Peak number; ^3^ t_R_—retention time; ^4^ ω—mass fraction of the component as area in % of the 100.00% of all identified peaks; 4 nd: not detected. *—components with content ≥ 0.040% wt. of extract are given, each value represents the mean ± SD (n = 3, *p* = 0.95).

**Table 5 plants-12-03401-t005:** Correlation between phytochemicals (group composition) and antioxidant activity of JB1 (Pearson’s coefficient (r)).

Antioxidant Parameters	Total Phenolic Compounds	Total Flavonoids	Total Terpenoids	Total Ascorbic acid
IC_50_ DPPH, A-*m* ^1^	−0.672	0.42	−0.659	−0.054
IC_50_ DPPH, A-*usm*	0.990 *	0.562	0.477	0.614
IC_50_ DPPH, M-*m*	0.628	0.909	0.659	0.845
IC_50_ DPPH, M-*usm*	* 0.925	0.985 *	0.837	0.315
IC_50_ DPPH, E-*m*	0.187	−0.332	0.078	0.075
IC_50_ DPPH, E-*usm*	0.172	0.223	0.208	0.262

^1^ A-*m*—Acetone, maceration; A-*usm*—acetone, ultrasound-assisted maceration; M-*m*—methanol, maceration; M- *usm*—methanol, ultrasound-assisted maceration; E-*m*—70% ethanol, maceration; E-*usm*—70% ethanol, ultrasound-assisted maceration; * significance (*p* < 0.05).

**Table 6 plants-12-03401-t006:** Correlation between phytochemicals (group composition) and antioxidant activity of JB2 berries (Pearson’s coefficient (r)).

Antioxidant Parameters	Total Phenolic Compounds	Total Flavonoids	Total Terpenoids	Total Ascorbic acid
IC_50_ DPPH, A-*m* ^1^	0.765	−0.688	−0.156	−0.377
IC_50_ DPPH, A-*usm*	0.241	−0.053	0.538	−0.978 *
IC_50_ DPPH, M-*m*	−0.501	−0.944 *	−0.827	−0.947 *
IC_50_ DPPH, M-*usm*	−0.36	−0.462	−0.524	−0.496
IC_50_ DPPH, E-*m*	−0.207	−0.153	0.177	0.076
IC_50_ DPPH, E-*usm*	0.558	0.357	0.522	0.439

^1^ A-*m*—Acetone, maceration; A-*usm*—acetone, ultrasound-assisted maceration; M-*m*—methanol, maceration; M-*usm*—methanol, ultrasound-assisted maceration; E-*m*—70% ethanol, maceration; E-*usm*—70% ethanol, ultrasound-assisted maceration; * significance (*p* < 0.05).

**Table 7 plants-12-03401-t007:** Antimicrobial activity of JB1 and JB2 extracts obtained by maceration and US with different solvents and redissolved in the solvent eutectic concentration of 1%.

Sample		*Clavibacter michiganensis*		*Erwinia carotovora* spp.		*Rhizoctonia solani*		*Alternaria solani*	
		MIC, μg/mL ^2^	MBC, μg/mL	MIC, μg/mL	MBC, μg/mL	MIC, μg/mL	MFC, μg/mL	MIC, μg/mL	MFC, μg/mL
JB1P-*m* ^1^		** 156 ± 13	** 312 ± 30	** 625 ± 45	1250 ± 100	* 312 ± 25	** 312 ± 25	** 625 ± 40	* 1250 ± 120
JB2P-*m*		** 1250 ± 120	** 1250 ± 100	** 1250 ± 100	>2500	* 2500 ± 135	** 2500 ± 170	** 2500 ± 180	* 2500 ± 180
JB1P-*usm*		* 312 ± 30	* 312 ± 30	* 312 ± 30	** 625 ± 45	* 312 ± 30	* 625 ± 30	* 625 ± 43	** 625 ± 40
JB2P-*usm*		* 625 ± 45	* 1250 ± 130	* 625 ± 45	** 1250 ± 110	* 1250 ± 100	* 1250 ± 100	* 1250 ± 100	** 1250 ± 100
JB1C-*m*		* 156 ± 15	* 156 ± 15	625 ± 45	* 625 ± 45	* 312 ± 30	625 ± 35	** 625 ± 40	1250 ± 100
JB2C-*m*		* 1250 ± 100	* 2500 ± 130	625 ± 45	* 1250 ± 100	* 2500 ± 125	>2500	** 1250 ± 100	>2500
JB1C-*usm*		* 312 ± 25	* 312 ± 30	** 312 ± 30	** 625 ± 50	* 625 ± 45	1250 ± 100	1250 ± 100	1250 ± 100
JB2C-*usm*		* 1250 ± 100	* 2500 ± 135	** 1250 ± 100	** 1250 ± 100	* 2500 ± 130	>2500	1250 ± 100	>2500
JB1A-*m*		** 39 ± 3	* 39 ± 5	** 39 ± 3	* 39 ± 4	* 156 ± 13	* 312 ± 25	* 312 ± 30	** 625 ± 30
JB2A-*m*		** 1250 ± 100	* 1250 ± 100	** 625 ± 50	* 1250 ± 100	* 625 ± 45	* 1250 ± 100	* 625 ± 35	** 1250 ± 100
JB1A-*usm*		* 39 ± 4	39 ± 3	** 39 ± 3	* 78 ± 7	* 39 ± 3	* 78 ± 5	** 39 ± 3	* 78 ± 5
JB2A-*usm*		* 2500 ± 170	>2500	** 1250 ± 100	* 2500 ± 130	* 1250 ± 100	* 2500 ± 180	** 625 ± 40	* 1250 ± 100
JB1M-*m*		* 156 ± 13	* 312 ± 25	1250 ± 100	2500 ± 125	* 156 ± 15	* 312 ± 25	* 156 ± 10	312 ± 25
JB2M-*m*		* 625 ± 45	* 1250 ± 100	1250 ± 100	>2500	* 625 ± 50	* 1250 ± 100	* 1250 ± 100	>2500
JB1M-*usm*		156 ± 13	156 ± 16	1250 ± 100	** 1250 ± 100	** 156 ± 13	* 312 ± 25	* 156 ± 10	2500 ± 180
JB2M-*usm*		156 ± 13	156 ± 13	1250 ± 100	** 2500 ± 150	** 625 ± 45	* 625 ± 45	* 1250 ± 100	2500 ± 175
JB1E-*m*		* 39 ± 3	* 78 ± 7	** 156 ± 15	312 ± 25	* 312 ± 25	625 ± 44	* 156 ± 12	312 ± 27
JB2E-*m*		* 1250 ± 100	* 2500 ± 155	** 2500 ± 160	>2500	* 2500 ± 155	>2500	* 1250 ± 100	1250 ± 100
JB1E-*usm*		* 78 ± 7	* 156 ± 15	** 312 ± 25	* 625 ± 50	* 156 ± 15	312 ± 27	78 ± 7	156 ± 15
JB2E-*usm*		* 625 ± 50	* 1250 ± 110	** 2500 ± 150	>2500	* 2500 ± 160	>2500	2500 ± 170	>2500
Positive Controls	α-pinene	156 ± 13	156 ± 13	625 ± 45	1250 ± 100	312 ± 30	625 ± 40	312 ± 30	625 ± 45
	Chloramphenicol ^3^	125 ± 12.3	625 ± 50	125 ± 12.3	125 ± 14	^5^ n.t.	n.t.	n.t.	n.t.
	Difenoconazole ^4^	n.t.	n.t.	n.t.	n.t.	3.9 ± 0.29	3.9 ± 0.29	1.9 ± 0.15	31.3 ± 3.1

^1^ JB1P-*m*—first year, pentane, maceration; JB2P-*m*—second year, pentane, maceration; JB1P-*usm*—first year, pentane, ultrasound-assisted maceration; JB2P-*usm*—second year, pentane, ultrasound-assisted maceration; JB1C-*m*—first year, chloroform, maceration; JB2C-*m*—second year, chloroform, maceration; JB1C-*usm*—first year, chloroform, ultrasound-assisted maceration; JB2C-*usm*—second year, chloroform, ultrasound-assisted maceration; JB1A-*m*—first year, acetone, maceration; JB2A-*m*—second year, acetone, maceration; JB1A-*usm*—first year, acetone, ultrasound-assisted maceration; JB2A-*usm*—second year, acetone, ultrasound-assisted maceration; JB1M-*m*—first year, methanol, maceration; JB2M-*m*—second year, methanol, maceration; JB1M-*usm*—first year, methanol, ultrasound-assisted maceration; JB2M-*usm*—second year, methanol, ultrasound-assisted maceration; JB1E-*m*—first year, 70% ethanol, maceration; JB2E-*m*—second year, 70% ethanol, maceration; JB1E-*usm*—first year, 70% ethanol, ultrasound-assisted maceration; JB2E-*usm*—second year, 70% ethanol, ultrasound-assisted maceration; ^2^ Each value represents mean ± SD; ^3^ The comparison drug for phytopathogenic bacteria strains; ^4^ The comparison drug for fungi strains; ^5^ n.t.—not tested; MIC—minimum inhibitory concentrations; MBC—minimum bactericidal concentrations; MFC—minimum fungicidal concentrations; * statistical differences at *p* < 0.05 probability level JB1 relative to JB2; **: statistical differences at *p* < 0.01 probability level JB1 relative to JB2.

**Table 8 plants-12-03401-t008:** Toxic effects of JB1 and JB2 extracts (extracted with acetone and 70% ethanol and redissolved at a solvent eutectic concentration of 1%) on *C. briggsae* and *C. elegans* organisms.

Sample	Concentration	^2^ The Percentage of Dead Nematodes, %	
		3 h	24 h
*Caenorhabditis elegans*			
propylene glycol (control)	0.05%	18.0 ± 3.8	42.0 ± 4.9
JB1A-*m* ^1^	0.05%	14.0 ± 2.5	96.5 ± 1.3
JB2A-*m*	0.05%	7.0 ± 1.8	20.0 ± 2.8
JB1E-*m*	0.05%	3.0 ± 1.2	91.5 ± 2.0
JB2E-*m*	0.05%	3.5 ± 1.3	31.5 ± 3.3
*Caenorhabditis briggsae*			
		3 h	24 h
propylene glycol	0.025%	5.0 ± 2.1	6.0 ± 2.3
propylene glycol	0.05%	18.0 ± 3.8	22.0 ± 4.1
JB1A-*m*	0.025%	0.5 ± 0.5	25.0 ± 3.1
JB1A-*m*	0.05%	17.5 ± 2.7	50.5 ± 3.5
JB2A-*m*	0.025%	8.5 ± 1.9	18.0 ± 2.7
JB2A-*m*	0.05%	11.0 ± 2.2	26.5 ± 3.1
JB1E-*m*	0.025%	9.5 ± 2.1	85.0 ± 2.5
JB1E-*m*	0.05%	19.0 ± 2.8	87.0 ± 2.4
JB2E-*m*	0.025%	8.5 ± 2.0	14.5 ± 2.5
JB2E-*m*	0.05%	14.0 ± 2.5	22.0 ± 2.9

^1^ JB1A-*m*—First year, acetone, maceration; JB2A-*m*—second year, acetone, maceration; JB1E-*m*—first year, 70% ethanol, maceration; JB2E-*m*—second year, 70% ethanol, maceration; ^2^ Each value represents mean ± SD.

**Table 9 plants-12-03401-t009:** Effect of acetone and 70% ethanol extracts of JB1 and JB2 (redissolved at a solvent eutectic concentration of 1%) on the resistance of *C. elegans* organisms to paraquat dichloride.

Sample, Solvent	Concentration	^2^ The Percentage of Dead Nematodes, %	
		without Paraquat Dichloride	0.5 mmol Paraquat Dichloride
propylene glycol (control)	0.05%	2.0 ± 1.1	20.0 ± 3.2
JB1A-*m* ^1^	0.05%	26.0 ± 3.5	29.3 ± 3.7
JB2A-*m*	0.05%	17.3 ± 3.0	14.7 ± 2.8
JB1E-*m*	0.05%	17.3 ± 3.0	69.3 ± 3.7
JB2E-*m*	0.05%	18.7 ± 3.1	26.7 ± 2.8

^1^ JB1A-*m*—First year, acetone, maceration; JB2A-*m*—second year, acetone, maceration; JB1E-*m*—first year, 70% ethanol, maceration; JB2E-*m*—second year, 70% ethanol, maceration; ^2^ Each value represents the mean ± SD.

## Data Availability

The data presented in this study are available upon request from the corresponding author.

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
