# Peer review of "Comparative Analysis of Chemical Profile and Biological Activity of *Juniperus communis* L. Berry Extracts"

_plants, 2023, doi:10.3390/plants12193401_

Round 1
Reviewer 1 Report
General comments:
1. Manuscript should be edited for English. There are a lot of small errors and inconsistencies.
2. Please label your samples (1st year , 2nd year) in some different and more simple way to be used in text. This construction "berries of maturity 1- and 2-years" is quite hard to follow and little bit confusing when it is written and used in some long sentence.
3. Since authors followed increased polarity of solvents to be applied for extraction why you did not used water, for instance, as solvent with the highest polarity? Since you use pentane as pure non-polar solvent? And on what basis you have choose 70% ethanol? Why not 80% or 50% or some other aqueous mixture? Also, one important observation related to this- this 70% EtOH" must be named as "ethanolic" extract not as "ethanol" since this was not pure alcohol but aqueous mixture. Please clarify/correct these important issues.
4. There is constant error in Tables, related to decimal numbers. Namely, all decimals should be given with point not with comma to be in the line with English grammar. Please carefully revise all Tables. In addition, related to results in Tables, why authors did not perform any statistical test (like t-test or Tuckey, etc.) to determine statistical significance of differences for results? In given way results are incomplete.
Specific comments
All are listed below with an appropriate Line number(s) from text in order to facilitate tracking:
Line 3: Latin name of plant in the title should be given in Italic. Correct.
Line 15: Always it should be "bioactive" not active. Please be consistent through a whole Manuscript and correct all errors related to this issue.
Line 17: Put this "against phytopathogens" in brackets here.
Lines 20-21: This other method should be labeled as "ultrasound-assisted maceration", not as you wrote. Please correct here as well as unify through a whole Manuscript.
Lines 24-26: Please rewrite given sentence. It is hard to follow and pretty much incomprehensible.
Line 27: First, it can not be "antioxidant activity profile". There are no profile for this. In addition, it "decreased" compared to what? Specify here.
Line 28: "was" not "is" here. It is result from your study so, it should be given in past tense.
Line 29: Strongly suggest to add "obtained" in front of "from" here.
Line 36: I think it should be here "pharmacy" not "pharmaceuticals" to be consistent with previous "chemistry" and "biochemistry".
Line 41: Latin name for genus here must be given in Italic. Correct.
Line 48: The same as previous.
Line 52: Delete "bioactive". It is just composition or chemical composition but it can not be bioactive composition.
Line 53: "phenolics" not "phenols". It is not the same in phytochemistry compared to classic Organic chemistry.
Line 63: Strongly suggest to replace "as high levels of..." with "can exhibit a high levels...". It seems to me much more appropriate in this context.
Lines 71-72: Please provide reference(s) for given statement.
Line 75: typo - Latin name of plant put in Italic here.
Lines 82-83: Please follow suggestion for rename of method given in Abstract section.
Line 93: Suggest to replace "certain" with "different" here.
Line 94: Well, it can not be "various" since you applied "only" two extraction techniques. So, simple use "two extraction techniques" here.
Line 95: I think it should be "are shown" here? Check/correct.
Lines 96-105: Can authors explain why/how is important to determine pH or electrical conductivity in phytochemical research? I have read a lot of phytochemical works and I have never seen that someone determine these two parameters for this type of study.
Lines 102-105: And what is now this "concentration of extracts" here? It is not specified in same way in the Table 1. Is this dry matter parameter actually? Please clarify this issue.
Line 108: "phenolics" as I explained earlier. Also, it should be "and" not "of terpenoids" here. Correct.
Lines 116-117: typos- no . after "dry". Delete.
Lines 118,121, 122 and 128: "phenolics". Correct.
Line 131: typo - put Latin name in Italic.
Line 132: typos- no capital letters for "sugars" and "flavonoids" in the names of assays in Table 2. Correct.
Lines 138-139: Here are we have opposite results trend to those presented in the Lines 128-130? Please check/clarify/correct.
Line 144: Did you mean "below limit of detection" here instead of "practically not identified"?
Line 147: typo - "Chromatographical" here. Correct.
Lines 148-150: The given statement is contrary to the statement made in the introduction of the paper. Where are reference(s) for given statement?
Line 151: Here we have only one reference for EOs (which is not even review article) so you can not speak about "the most studied".
Line 157: Suggest to rewrite as follow: "... of acetone and ethanolic extracts obtained from..."
Line 167: Please, "70% EtOH" is not absolute ethanol. Absolute must be 96%. Correct this.
Line 174: Latin name of plant put in Italic.
Line 176: Give term "delta" as Greek letter in the Table 3, not as a word. Also, for compound no. 37 in the Table 3 you must specify on what position is methyl group? If this is an acid. Otherwise, this is not acid but ester. Check/clarify/correct.
Line 184: Again, 70% EtOH is not an absolute. Correct.
Lines 191-209: And where are results for other solvents except acetone? Why they are not presented in full form like this?
Line 199: Please do not use term "primary metabolites" here since you determine only soluble sugars i.e. monosaccharides here.
Line 202: Greek symbol for delta here, not word. Correct.
Line 208: typo - "acetol" not "-acetol" here. Correct.
Lines 211-214: Your sentence here is incomplete. Missing some verb construction at the end in the Line 214. Check/correct.
Line 215: "yielded" here instead of "identified". Solvent can not identify compounds.
Line 216: Greek letter for gamma here instead of word. Correct.
Line 220: incorrect subtitle. DPPH is a radical and it is quenched not bind by phenolics and other similar antioxidants. Correct subtitle and terminology used.
Line 221: Latin name in Italic here. Correct.
Lines 222-223: Repeating of already said.
Line 225: Again, incorrect terminology. It can not be "abosrbing" but "neutralizing" or "quenching". Correct.
Lines 238-240: On what basis authors choose alpha-pinene here? Is it some positive control or what? It was not mentioned anywhere in text. Also, it is not specified in Experimental section in description of DPPH radical quenching assay. Please correct/clarify.
Lines 248-249: Sentence must be rewritten. Suggest as follow:" Among applied extractants pentane and chloroform have been confirmed as the least efficient for bioactive compounds extraction. Possible reason for this can be low solvents polarity."
Line 250: Delete this "The antioxidant potential of the". It is surplus here.
Lines 252-253: incomprehensible text. What are these "phenolic hydroxyls" here? How it is related to higher content of phenolic compounds?? Please rewrite a whole paragraph to become understandable.
Line 254: "was the highest" instead of "were mainly found" here. Also, what are "metal extracts" mentioned here???
Lines 269,270,272: Correct plant Latin name to be italicized.
Lines 276-277: incomplete sentence. Missing something like "be used as positive control" at the end of sentence for alpha-pinene here.
Line 280: Add "and" in front of "alternariosis" here.
Line 284: I do not understand this "1% /micrograms/mL". How 1% can be the same as this unit?? Please check/correct/clarify. In addition please provide unit for MIC/MBC results in the Table 4.
Lines 302-303: Fully incomprehensible explanation. Please rewrite a whole part of text here.
Lines 317-318: Please provide reference(s) for given statement.
Lines 349-350: typo - put Latin name in Italic here.
Line 358: Put Latin name in Italic here. From here, this error is repeating (for instance Lines 360, 363, 369, 370, 371, 373, etc.). Please check a rest of text and put all Latin names in Italic.
Line 391: "phenolics" not "phenols".
Line 392: "by using" not "when using".
Line 394: "the highest" here. Correct.
Line 397: "bioactive". Correct.
Line 403: "general" not "total" here. Correct.
Line 410: Please there are no such thing like "biochemical" composition. It can be chemical or in your case more precise "phytochemical". Correct.
Lines 411-414: Please provide reference(s) for given statements.
Lines 415-422 and Lines 424-432: Here we have repeating of one same statement/results? Please check/remove surplus. Also, sentence finishing in the Line 423 is incomplete. Missing some verb at the end. Add.
Lines 418, 420: typos- Latin name in Italic, split next sentence from previous with space.
Line 437: The listed compounds are also secondary metabolites so the given statement is in contrast to one given int he Lines 433-434. Check and unify to be consistent.
Line 438: Please it can not be "The analysis of DPPH" but " The analyisis of DPPH radical assays results showed..." Correct.
Line 439: "the highest" not "the greatest". Correct.
Line 440: As I explained earlier there are no such things like "profile of antioxidant activity". Please rewrite.
Line 442: What is this "reactions of intermolecular interaction"?? It is meaningless. Please rewrite.
Lines 447-450: Incomprehensible text. Please rewrite and clarify.
Lines 451-452: incomprehensible "and was only of antioxidant nature". What is this?? Please clarify/rewrite.
Line 468: Again, you can not state that something is not scarce in literature with only one reference like here.
Line 470: "J. communis L." here. Correct. Same in the Line 475.
Line 477: typo - no . after Latin name "S. aureus" here. Correct.
Lines 504 and 507: typos - Put Latin name of genus in Italic here.
Line 553: "dried extracts were". Correct.
Lines 571-591: Please reorder- fist to be total phenolic and than total flavonoid content.
Line 582: Please "total phenolic content" not "total phenolic number". There are no any numbers in this assay title.
Lines 592-597: Strongly suggest to rename this as "total soluble sugars". Because you did not determine all sugars in this assay. Only those that were soluble under applied conditions and solvents.
Line 662: incorrect terminology. Please, DPPH radical assay is not assay used for total antioxidant capacity determination. It just present ability to quench free radicals. Total antioxidant capacity (TAC) is related to in vitro phosphomolybdenum assay known in the literature: https://doi.org/10.1006/abio.1999.4019
Lines 769-807: Please make conclusion more concise and shorten. Also, apply new terminology suggested through Manuscript to be consistent.
Kind regards.
A whole text should be checked by some professional lector or English native speaker in order to make it fluid and fully understandable. There are a lot of small errors making hard to follow main idea of text.
Author Response
Comments 1. Manuscript should be edited for English. There are a lot of small errors and inconsistencies.
Response: Corrected.
Comments 2. Please label your samples (1st year, 2nd year) in some different and more simple way to be used in text. This construction "berries of maturity 1- and 2-years" is quite hard to follow and little bit confusing when it is written and used in some long sentence.
Response: Done. 1st year, 2nd year use in article
Comments 3. Since authors followed increased polarity of solvents to be applied for extraction why you did not used water, for instance, as solvent with the highest polarity? Since you use pentane as pure non-polar solvent? And on what basis you have choose 70% ethanol? Why not 80% or 50% or some other aqueous mixture? Also, one important observation related to this- this 70% EtOH" must be named as "ethanolic" extract not as "ethanol" since this was not pure alcohol but aqueous mixture. Please clarify/correct these important issues.
Response:
1) Despite the fact that water is one of the most commonly used extractants and has a number of advantages (pharmacologically indifferent, non-flammable and non-explosive) it has many disadvantages: -does not dissolve or extract hydrophobic substances; -does not possess antiseptic properties, as a result of which microorganisms may develop in aqueous extracts. microorganisms can develop in aqueous extracts, which can cause spoilage of the product; - water causes hydrolytic cleavage of many substances, especially at high temperature; it does not have antiseptic properties. at high temperature;
2) The 70% ethanol used in the study was the best extractant according to other studies due to its lower toxicity and increased extraction of polyphenols with lower selectivity. Another study reported a more preferable choice of 70% ethanol for the isolation of bioactive compounds.
Fejér, J.; Gruľová, D.; Eliašová, A.; Kron, I. Seasonal Variability of Juniperus communis L. Berry Ethanol Extracts: 2. In Vitro Ferric Reducing Ability of Plasma (FRAP) Assay. Molecules 2022, 27, 9027. https://doi.org/10.3390/molecules27249027
Wieczorek, P.P.; Hudz, N.; Yezerska, O.; Horčinová-Sedláčková, V.; Shanaida, M.; Korytniuk, O.; Jasicka-Misiak, I. Chemical Variability and Pharmacological Potential of Propolis as a Source for the Development of New Pharmaceutical Products. Molecules 2022, 27, 1600. https://doi.org/10.3390/molecules27051600
3) Pentane as a representative of non-polar extractants extracts more than 97% of lipophilic compounds, which may also exhibit a potentially high biological profile. Among the compounds extracted by pentane, possessing high biological activity are aglycones of cardiac glycosides, bases of bases of most alkaloids, flavones, sapogenins, waxes.
4) Done/ correct (ethanolic)
Comments 4. There is constant error in Tables, related to decimal numbers. Namely, all decimals should be given with point not with comma to be in the line with English grammar. Please carefully revise all Tables. In addition, related to results in Tables, why authors did not perform any statistical test (like t-test or Tuckey, etc.) to determine statistical significance of differences for results? In given way results are incomplete.
Response: Thank you for pointing this out. We agree with this comment. Done, lines 874-88. Added the following chapter 4.7. Statistical Analysis
The results were summarized as the mean values ± standard deviation (SD). The data were processed using Microsoft Excel 2016 and OriginPro 9.5 (OriginLab Co., Northampton, MA, USA). Data were compared using a non-parametric Kruskal-Wallis test. The precise p-values were calculated for the pair-wise comparisons between the groups using the Mann-Whitney test. Data analysis employed IBM SPSS 23.0 statistical software (Chicago, IL, USA). The level of p < 0.05 was considered statistically signifi-cant. Pearson correlations (p < 0.05) were used to identify correlations between phyto-chemical constituents and antioxidant activity of acetone, methanol and 70% ethanolic extracts of juniper berries (J. communis L.) 1st and 2- years of maturity.
Statistical processing of the results of nematocidal activity was carried out using the Fisher angular transformation. Regression analysis of calibration characteristics and mathematical prediction of the peak area response for chromatographic analysis were performed using OriginPro 9.5 software (Origin Lab Corp, Northampton, MA, USA) and processed in accordance with the standards [93].
Specific comments
Comments Line 3: Latin name of plant in the title should be given in Italic. Correct.
Response: Corrected
Comments Line 15: Always it should be "bioactive" not active. Please be consistent through a whole Manuscript and correct all errors related to this issue.
Response: Corrected
Comments Line 17: Put this "against phytopathogens" in brackets here.
Response: Corrected
Comments Lines 20-21: This other method should be labeled as "ultrasound-assisted maceration", not as you wrote. Please correct here as well as unify through a whole Manuscript.
Response: Thank you for pointing this out. We agree with this comment. Corrected
Comments Lines 24-26: Please rewrite given sentence. It is hard to follow and pretty much incomprehensible.
Response: Corrected.
“Antimicrobial activity was higher in berries of 1 year maturity, while acetone extract during ultrasound-assisted maceration was the most bioactive in relation to the phytopathogens.”
Comments Line 27: First, it can not be "antioxidant activity profile". There are no profile for this. In addition, it "decreased" compared to what? Specify here.
Response: Corrected, lines 25-27
Depending on the extraction method and the choice of solvent, the antioxidant activity with the use of UAM decreased by 1.5-1.9 times compared to the extracts obtained by maceration.
Comments Line 28: "was" not "is" here. It is result from your study so, it should be given in past tense.
Response: Corrected
Comments Line 29: Strongly suggest to add "obtained" in front of "from" here.
Response: Corrected
Comments Line 36: I think it should be here "pharmacy" not "pharmaceuticals" to be consistent with previous "chemistry" and "biochemistry".
Response: Corrected
Comments Line 41: Latin name for genus here must be given in Italic. Correct.
Response: Corrected
Comments Line 48: The same as previous.
Response: Corrected
Comments Line 52: Delete "bioactive". It is just composition or chemical composition but it can not be bioactive composition.
Response: Corrected
Comments Line 53: "phenolics" not "phenols". It is not the same in phytochemistry compared to classic Organic chemistry.
Response: Corrected
Comments Line 63: Strongly suggest to replace "as high levels of..." with "can exhibit a high levels...". It seems to me much more appropriate in this context.
Response: Replaced, line 67
Comments Lines 71-72: Please provide reference(s) for given statement.
Response: Done, line 86
Comments Line 75: typo - Latin name of plant put in Italic here.
Response: Corrected
Comments Lines 82-83: Please follow suggestion for rename of method given in Abstract section.
Response: Corrected
Comments Line 93: Suggest to replace "certain" with "different" here.
Response: Corrected
Comments Line 94: Well, it can not be "various" since you applied "only" two extraction techniques. So, simple use "two extraction techniques" here.
Response: Corrected
Comments Line 95: I think it should be "are shown" here? Check/correct.
Response: Corrected
Comments Lines 96-105: Can authors explain why/how is important to determine pH or electrical conductivity in phytochemical research? I have read a lot of phytochemical works and I have never seen that someone determine these two parameters for this type of study.
Determination of pH and electrical conductivity of the extracts are given in the study because they affect the application efficiency and accelerate the decomposition of pesticides. In addition, the dissociation constant of many chemicals depends on pH, which further affects the uptake of active substances by plant tissues when the extract is used in phytopathogen control [1, 2].
1) Reffstrup, T. K.; Larsen, J. C.; Meyer, O. Risk assessment of mixtures of pesticides. Current approaches and future strategies. Regulatory Toxicology and Pharmacology. 2010, 56 (2), 174-192
Physicochemical parameters can influence the biological characteristics of insecticides and their activity. Determination of pH and electrical conductivity is important to determine the possibility of use as insecticidal preparations or in conjunction with industrial chemical pesticides.
2) Vechia, J.F.D.; Andrade, D.J.; Azevedo, R.G.; Azevedo, M.C. Effects of insecticide and acaricide mixtures on Diaphorina citri control. Plant Protection. 2019, 41 (1). DOI: https://doi:10.1590/0100-29452019076
Comments Lines 102-105: And what is now this "concentration of extracts" here? It is not specified in same way in the Table 1. Is this dry matter parameter actually? Please clarify this issue.
Response: Corrected.
“ Extraction yield (%)”
Comments Line 108: "phenolics" as I explained earlier. Also, it should be "and" not "of terpenoids" here. Correct.
Response: Corrected.
Comments Lines 116-117: typos- no . after "dry". Delete.
Response: Corrected.
Comments Lines 118,121, 122 and 128: "phenolics". Correct.
Response: Corrected.
Comments Line 131: typo - put Latin name in Italic.
Response: Corrected.
Comments Line 132: typos- no capital letters for "sugars" and "flavonoids" in the names of assays in Table 2. Correct.
Response: Corrected.
Comments Lines 138-139: Here are we have opposite results trend to those presented in the Lines 128-130? Please check/clarify/correct.
Response: Corrected.
Comments Line 144: Did you mean "below limit of detection" here instead of "practically not identified"?
Response: Corrected.
Comments Line 147: typo - "Chromatographical" here. Correct.
Response: Corrected.
Comments Lines 148-150: The given statement is contrary to the statement made in the introduction of the paper. Where are reference(s) for given statement?
Response: Corrected, lines 449-454
Comments Line 151: Here we have only one reference for EOs (which is not even review article) so you can not speak about "the most studied".
Response: corrected, line 449-451.
Currently, a large array of data has been accumulated on the chemical composition of essential oils of juniper berries (J. communis L.) [3,40], while the composition of extracts is given little attention in available literature [28, 30, 41].
Comments Line 157: Suggest to rewrite as follow: "... of acetone and ethanolic extracts obtained from..."
Response: Corrected
Comments Line 167: Please, "70% EtOH" is not absolute ethanol. Absolute must be 96%. Correct this.
Response: Corrected
Comments Line 174: Latin name of plant put in Italic.
Response: Corrected
Comments Line 176: Give term "delta" as Greek letter in the Table 3, not as a word. Also, for compound no. 37 in the Table 3 you must specify on what position is methyl group? If this is an acid. Otherwise, this is not acid but ester. Check/clarify/correct.
Response: Corrected
Comments Line 184: Again, 70% EtOH is not an absolute. Correct.
Response: Corrected
Comments Lines 191-209: And where are results for other solvents except acetone? Why they are not presented in full form like this?
Response: In this work, 70% ethanol and acetone extracts showed higher values of antimicrobial and antioxidant activities and were further investigated for individual chemical composition
Comments Line 199: Please do not use term "primary metabolites" here since you determine only soluble sugars i.e. monosaccharides here.
Response: Corrected
Comments Line 202: Greek symbol for delta here, not word. Correct.
Response: Corrected
Comments Line 208: typo - "acetol" not "-acetol" here. Correct.
Response: Corrected
Comments Lines 211-214: Your sentence here is incomplete. Missing some verb construction at the end in the Line 214. Check/correct.
Response: Corrected
in the case of the acetone extract of the berries of the 1st year, bornyl acetate and dihydroabietinol were additionally identified
Comments Line 215: "yielded" here instead of "identified". Solvent cannot identify compounds.
Response: Corrected
Comments Line 216: Greek letter for gamma here instead of word. Correct.
Response: Corrected
Comments Line 220: incorrect subtitle. DPPH is a radical and it is quenched not bind by phenolics and other similar antioxidants. Correct subtitle and terminology used.
Response: Corrected – “Analysis of the extracts’ effect on DPPH-scavenging activity”
Comments Line 221: Latin name in Italic here. Correct.
Response: Corrected
Comments Lines 222-223: Repeating of already said.
Response: Corrected
Comments Line 225: Again, incorrect terminology. It can not be "abosrbing" but "neutralizing" or "quenching". Correct.
Response: Corrected – “The neutralizing effects”
Comments Lines 238-240: On what basis authors choose alpha-pinene here? Is it some positive control or what? It was not mentioned anywhere in text. Also, it is not specified in Experimental section in description of DPPH radical quenching assay. Please correct/clarify.
Response: Corrected
In this work, alpha-pinene, which is a major component of juniper berries, was used as a positive control for the antimicrobial method (Table 6).
Comments Lines 248-249: Sentence must be rewritten. Suggest as follow:" Among applied extractants pentane and chloroform have been confirmed as the least efficient for bioactive compounds extraction. Possible reason for this can be low solvents polarity."
Response: Thank you for pointing this out. We changed it and emphasize this point, lines 240-241
Comments Line 250: Delete this "The antioxidant potential of the". It is surplus here.
Response: Corrected
Comments Lines 252-253: incomprehensible text. What are these "phenolic hydroxyls" here? How it is related to higher content of phenolic compounds?? Please rewrite a whole paragraph to become understandable.
Response: Corrected
The extracts from berries of 1st year had a more pronounced antioxidant activity compared to berries of 2nd year; which may indicate a higher content of phenolic compounds, the total content of monoterpenes, sesquiterpenes and other biologically active components, which were the highest in 70% ethanolic, acetone and methanol extracts.
Comments Line 254: "was the highest" instead of "were mainly found" here. Also, what are "metal extracts" mentioned here???
Response: Corrected
Comments Lines 269,270,272: Correct plant Latin name to be italicized.
Response: Corrected
Comments Lines 276-277: incomplete sentence. Missing something like "be used as positive control" at the end of sentence for alpha-pinene here.
Response: Corrected, lines 818-820
As an additional comparison drug, a natural antimicrobial agent – α-pinene be used as positive control, which was the major component in berry extracts according to GC/MS data analysis.
Comments Line 280: Add "and" in front of "alternariosis" here.
Response: Corrected
Comments Line 284: I do not understand this "1% /micrograms/mL". How 1% can be the same as this unit?? Please check/correct/clarify. In addition please provide unit for MIC/MBC results in the Table 4.
Response: Corrected, lines 611-613: «dried extracts were re-dissolved (to obtain a 1% solution of extracts) in a solvent eutectic consisting of 65% 1,3-propylene glycol, 25% ethanol and 10% water. »
Comments Lines 302-303: Fully incomprehensible explanation. Please rewrite a whole part of text here.
Response: Done, lines 313-314
The use of ultrasound for 1st year and 2nd year berry extracts in case of acetone, methanol and pentane leads to an increase in antimicrobial activity, in case of 70% ethanolic the antimicrobial activity decreases.
Comments Lines 317-318: Please provide reference(s) for given statement.
Response: added reference
Comments Lines 349-350: typo - put Latin name in Italic here.
Response: Corrected
Comments Line 358: Put Latin name in Italic here. From here, this error is repeating (for instance Lines 360, 363, 369, 370, 371, 373, etc.). Please check a rest of text and put all Latin names in Italic.
Response: Corrected
Comments Line 391: "phenolics" not "phenols".
Response: Corrected
Comments Line 392: "by using" not "when using".
Response: Corrected
Comments Line 394: "the highest" here. Correct.
Response: Corrected
Comments Line 397: "bioactive". Correct.
Response: Corrected
Comments Line 403: "general" not "total" here. Correct.
Response: Corrected
Comments Line 410: Please there are no such thing like "biochemical" composition. It can be chemical or in your case more precise "phytochemical". Correct.
Response: Corrected
Comments Lines 411-414: Please provide reference(s) for given statements.
Response: Corrected
Comments Lines 415-422 and Lines 424-432: Here we have repeating of one same statement/results? Please check/remove surplus. Also, sentence finishing in the Line 423 is incomplete. Missing some verb at the end. Add.
Response: Done, lines 470-473
For comparison, Elboughdiri et al [46] found that analysis of 2nd year berry ethanol extract showed that the composition was dominated by monoterpenes (α-pinene: 39.12%, sabinene: 8.87%, β-pinene: 12.68%, myrcene: 12.92%, limonene: 2.23%) and sesquiterpenes (β-caryophyllene: 4.41%, α-humulene: 1.05%, germacrene D: 4.23%, δ-cadinene: 1.35%)
Comments Lines 418, 420: typos- Latin name in Italic, split next sentence from previous with space.
Response: Corrected
Comments Line 437: The listed compounds are also secondary metabolites so the given statement is in contrast to one given int he Lines 433-434. Check and unify to be consistent.
Response: Corrected
Comments Line 438: Please it can not be "The analysis of DPPH" but " The analyisis of DPPH radical assays results showed..." Correct.
Response: Corrected, line 490
Comments Line 439: "the highest" not "the greatest". Correct.
Response: Corrected
Comments Line 440: As I explained earlier there are no such things like "profile of antioxidant activity". Please rewrite.
Response: Corrected, lines 492-495
Depending on the extraction method, the antioxidant activity decreased by 1.5-1.9 times with the use of ultrasound, which may be due to cavitation processes when using ultrasound and the reactions of intermolecular interaction occurring at the same time.
Comments Lines 447-450: Incomprehensible text. Please rewrite and clarify.
Response: Corrected, lines 498-500
Comments Lines 451-452: incomprehensible "and was only of antioxidant nature". What is this?? Please clarify/rewrite.
Response: Corrected, lines 501-503
Comments Line 468: Again, you can not state that something is not scarce in literature with only one reference like here.
Response: added references, lines 81-84
Comments Line 470: "J. communis L." here. Correct. Same in the Line 475.
Response: Corrected
Comments Line 477: typo - no . after Latin name "S. aureus" here. Correct.
Response: Corrected
Comments Lines 504 and 507: typos - Put Latin name of genus in Italic here.
Response: Corrected
Comments Line 553: "dried extracts were". Correct.
Response: Corrected
Comments Lines 571-591: Please reorder- fist to be total phenolic and than total flavonoid content.
Response: Corrected
Comments Line 582: Please "total phenolic content" not "total phenolic number". There are no any numbers in this assay title.
Response: Corrected
Comments Lines 592-597: Strongly suggest to rename this as "total soluble sugars". Because you did not determine all sugars in this assay. Only those that were soluble under applied conditions and solvents.
Response: Corrected
Comments Line 662: incorrect terminology. Please, DPPH radical assay is not assay used for total antioxidant capacity determination. It just present ability to quench free radicals. Total antioxidant capacity (TAC) is related to in vitro phosphomolybdenum assay known in the literature: https://doi.org/10.1006/abio.1999.4019
Response: Corrected
The assessment of the antioxidant activity is very specific to the sample and method, since the reaction mechanisms of the methods for determining antioxidant activity are completely different in terms of the types of oxidants, reaction conditions and even results.
Comments Lines 769-807: Please make conclusion more concise and shorten. Also, apply new terminology suggested through Manuscript to be consistent.
Response: Uptade, conclusion reduced.
The presented investigation shows the results group composition and bioactivity of the extracts of juniper berries (J. communis L.) 1-and 2- years of maturity collected in the temperate continental zone of the Republic of Tatarstan, the Russian Federation use various solvents (pentane, chloroform, acetone, methanol and 70% ethanolic) and extraction methods – maceration and ultrasound-assisted maceration. The group composition of berry extracts of 1 and 2 years of maturity, obtained with the help of different extractants and conditions, differed in the quantitative composition. The identified differences in the phytopathogenic activity of extracts depended on the choice of extractants and maceration conditions. The highest antioxidant and antimi-crobial activity corresponded to acetone and 70% ethanolic extract of berries 1 year of maturity.
Further, the individual chemical composition and nematocidal activity was de-termined for 70% ethanolic and acetone extract of berries which before showed higher biological profile. A total of 38 individual compounds were identified, depending on the selected extractant (24 compounds in berries of 2nd year, 23 compounds in berries of 1st year). According to GC-MS data, the main major compounds were α-pinene (20.6-42.3%), dihydroxyacetone (0.6-39.8%), myo-inositol (4.8-19.1%), germacrene D (4.1-14.8%), and β-myrcene (3.6-8.5%). The data of nematocidal activity indicated the presence of substances toxic to soil nematodes, especially for C. elegans, in the extracts of berries of 1 year maturity, and in the presence of paraquat, the activity in-creased several times.
According to the results of this study, the 70% ethanolic and acetone extracts of berries of 1st year of J. communis L. suggesting that they can be alternatives to conven-tional synthetic pesticides and could be used in the development of biopesticide prod-ucts after additional studies necessary, including further identification of components responsible for high phytopathogenic activity.
Reviewer 2 Report
In their MS plants-2579648, the authors studied the chemical composition and biological activity of various extracts of juniper berries (of maturity 1- and 2- years) in different solvents (pentane, chloroform, acetone, methanol, and 70% ethanol) obtained by two different methods (maceration and maceration with ultrasound). It is an extensive and interesting study, and the reviewer highly appreciates the authors' efforts. The MS has 76 references, 34 published in the last five years.
The following comments and suggestions are available below:
1. Introduction:
In this section, the data presented are mixed.
The authors are encouraged to organize the data presented in the following order: 1. systematic data (Family, Genus, Species); 2. Habitat; 3. Chemical composition; 4. Bioactivities (general presentation, not only regarding those presented in the present study); 5. therapeutical considerations in traditional medicine.
Then, in the introduction section's final paragraph, they could refer to the most recent studies from the scientific literature regarding Juniperus extracts, thus evidencing their study's novelty.
2. Materials and Methods
2.1. The entire study has no Data Analysis.
The authors are invited to perform Anova or t-test to show the statistically significant differences (p < 0.05) between extracts for all determinations.
They are also encouraged to show the correlations between phytoconstituents and biological activities (Pearson correlation or other correlation tests).
2.2. Subsections 4.1. and 4.2. could be reunited in a single subsection entitled "Materials."
2.3. Figure 4 is incomplete; the image with Conifers with berries of 2 years of maturity J. communis L. is missing.
2.4. In Extracts preparation, the authors could have marked two sub-subsections corresponding to maceration and maceration with ultrasounds for better differentiation and understanding.
2.4. The first three determinations (lines 565-570) are not a part of the Chemical composition analysis. Please, check and correct.
2.5. The authors are encouraged to mention as a separate subsection title for each performed analysis in the following order:
Total phenolic compounds content (not phenolic number as in line 582)
Total flavonoids
Total terpenoids
Ascorbic acid content
Sugar content
GC-MS Analysis
2.5. The same comment is available for biological activity determinations.
Moreover, the authors should indicate the following data:
- the GPS coordinates of the harvesting zone and the registering number of voucher specimens of row material.
- the provider of organisms used for biological studies or how the authors purchased them.
3. Results
3.1. For better understanding and maintaining the reader's interest, the analysis from this section should have the same titles in the same order as in Materials and Methods. The authors are encouraged to check and rectify this.
3.2. The authors should check and revise the following expressions: 1. Total phenolic compounds per gallic acid equivalent (GAE); 2. Total flavonoids per quercetin 132 equivalent (Que); 3. Total terpenoids per linalool equivalent (Lin). They are mentioned correctly in Materials and Methods in the following lines: 590-591, 580-581. The phrase from 614-615 should be checked and corrected accordingly.
3.3. Each Table should be revised. The authors are encouraged to register each value as a mean ± SD and mark the statistically significant values with superscripts. Moreover, in each Table footer, all abbreviations used should be explained.
3.4. For better understanding, the parameters from column 4 of Table 3 could be removed and detailed in Supplementary Material. Thus, only the main values remain, and this table's data are clearer.
3.5. The authors are encouraged to display the results of antioxidant activity determinations in correspondent tables, expressed as a mean ± SD, and mark the statistically significant values. Figures 1-3, without error bars and p-values, are irrelevant.
3.6. The final subsection of Results could be entitled Data Analysis; the authors should analyze the correlation between all quantified constituents and the variable parameters from bioactivities determinations.
The obtained correlations should be analyzed and compared in Discussion Section using suitable references.
4. Discussion
In this section, as an overview in a final paragraph, the authors should include a great part of the Conclusions.
5. Conclusions
In the current version, they are too long.
The authors are encouraged to maintain a few briefly presented ideas that deserve to be highlighted at the final of their study. Moreover, they could mention further directions or applications.
Author Response
- Introduction:
Comments. The authors are encouraged to organize the data presented in the following order: 1. systematic data (Family, Genus, Species); 2. Habitat; 3. Chemical composition; 4. Bioactivities (general presentation, not only regarding those presented in the present study); 5. therapeutical considerations in traditional medicine.
Response: Thank you for pointing this out. We agree with this comment. We changed it and emphasize this point. Done, lines 41-63.
The genus Juniperus is one of the most diverse among conifers and includes more than 60 species. Juniper (Juniperus communis L.) is an evergreen shrub of the genus Ju-niperus of the Cypress family, growing in the forests of Russia and forest—steppe zones of parts of Europe, Western and partly Eastern Siberia. Its population is spread globally, being the only Juniperus species found in both hemispheres, with reports of this plant in Arctic regions of Asia and North America [4]. These are dicotyledonous trees that produce seeds every 2 years, which have a circular or spherical shape [5]. In Ta-tarstan, it is most widespread on the territory of the Volga-Kama State Natural Bio-sphere Reserve (55°18′10″N 49°17′10″E) and the territories adjacent to it [6, 7]
It has been established that the common juniper berries contain a wide range of biologically active compounds: α-pinene, camphene, pectin, organic acids (glycolic, malic, ascorbic), cyclohexitol, terpenes, proteins, fermentable sugars, wax, gum, ca-dinene, juniper camphor [8]. To date, several reports have highlighted its antimicrobi-al, antifungal, antioxidant, anti-inflammatory, and antidiabetic potential, as well as its anticarcinogenic, hepatoprotective, neuronal, and renal effects [2, 9, 10]. Due to this, essential oils, extracts, biologically active fractions and individual compounds from juniper berries can be useful for the development of new pharmacological drugs in the treatment of a number of acute and chronic human diseases.
All types of junipers are characterized by a high content of essential oils and phenolic compounds and are largely used in folk medicine in various countries, showing a wide range of biological activity and industrial applications [11]. It is known that juniper has diuretic, anti-inflammatory, antifungal, analgesic, hepatoprotective, antimicrobial and antioxidant effects in traditional medicine [12].
Comments.Then, in the introduction section's final paragraph, they could refer to the most recent studies from the scientific literature regarding Juniperus extracts, thus evidencing their study's novelty.
Response: Thank you for pointing this out. We changed it and emphasize this point. Done, lines 75-96
- Materials and Methods
2.1. The entire study has no Data Analysis.
Comments. The authors are invited to perform Anova or t-test to show the statistically significant differences (p < 0.05) between extracts for all determinations.
Response: Thank you for pointing this out. We agree with this comment. Done, lines 874-888. Added the following chapter 4.7. Statistical Analysis
The results were summarized as the mean values ± standard deviation (SD). The data were processed using Microsoft Excel 2016 and OriginPro 9.5 (OriginLab Co., Northampton, MA, USA). Data were compared using a non-parametric Kruskal-Wallis test. The precise p-values were calculated for the pair-wise comparisons between the groups using the Mann-Whitney test. Data analysis employed IBM SPSS 23.0 statistical software (Chicago, IL, USA). The level of p < 0.05 was considered statistically signifi-cant. Pearson correlations (p < 0.05) were used to identify correlations between phyto-chemical constituents and antioxidant activity of acetone, methanol and 70% ethanolic extracts of juniper berries (J. communis L.) 1st and 2- years of maturity.
Statistical processing of the results of nematocidal activity was carried out using the Fisher angular transformation. Regression analysis of calibration characteristics and mathematical prediction of the peak area response for chromatographic analysis were performed using OriginPro 9.5 software (Origin Lab Corp, Northampton, MA, USA) and processed in accordance with the standards [93].
Comments. They are also encouraged to show the correlations between phytoconstituents and biological activities (Pearson correlation or other correlation tests).
Response: Thank you for pointing this out. We agree with this comment. Done, lines 249-268. Also included Pearson correlation – table:
Table 4. Correlation between phytochemical constituents (group composition) and antioxidant activity of berries of 1st year of maturity (Pearson's coefficient (r)).
Table 5. Correlation between phytochemical constituents (group composition) and antioxidant activity of berries of 2nd year of maturity (Pearson's coefficient (r)).
Comments. 2.2. Subsections 4.1. and 4.2. could be reunited in a single subsection entitled "Materials."
Response: Thank you for pointing this out. We agree with this comment. Done, line 562
Comments. 2.3. Figure 4 is incomplete; the image with Conifers with berries of 2 years of maturity J. communis L. is missing.
Response: Thank you for pointing this out. We agree with this comment. Done, line 584 - Figure 4. Conifers with berries of 1 year (A) and 2-year (B) of maturity J. communis L.
Comments. 2.4. In Extracts preparation, the authors could have marked two sub-subsections corresponding to maceration and maceration with ultrasounds for better differentiation and understanding.
Response: Done, lines 592, 611. Added the following chapters: 4.2. Extract Preparation by maceration and 4.3. Extract Preparation by ultrasound-assisted maceration
Comments. 2.4. The first three determinations (lines 565-570) are not a part of the Chemical composition analysis. Please, check and correct.
Response: Thank you for pointing this out. We agree with this comment. Done, line 620. Added the following chapters: 4.4. Physical parameters of extracts
Comments. 2.5. The authors are encouraged to mention as a separate subsection title for each performed analysis in the following order:
Total phenolic compounds content (not phenolic number as in line 582)
Response: Done, line 630. Added the following chapter: 4.5.1. Total phenolic compounds content
Total flavonoids
Response: Done, line 641. Added the following chapter: 4.5.2. The total content of flavonoids
Total terpenoids
Response: Done, line 653. Added the following chapter: 4.5.3. The total concentration of terpenoids
Ascorbic acid content
Response: Done, line 664. Added the following chapter: 4.5.4. The ascorbic acid content
Sugar content
Response: Done, line 677. Added the following chapter: 4.5.5. The sugar contents
GC-MS Analysis
Response: Done, line 692. Added the following chapter: 4.5.6. Chromatographic analysis of J. communis L. berry extracts
Comments. 2.5. The same comment is available for biological activity determinations.
Response: Done, lines 733, 734, 766, 829
Moreover, the authors should indicate the following data:
- the GPS coordinates of the harvesting zone and the registering number of voucher specimens of row material.
Response: Done, lines 570-571. Vysokogorsky district of Tatarstan (55.953357° N 49.183376° E).
- the provider of organisms used for biological studies or how the authors purchased them.
Response: Done, lines 767-771. Clavibacter michiganensis gram-positive strain VKM Ac-1404 (VKM IBFM RAS, Push-chino, Russia), gram-negative strain of bacteria Erwinia carotovora spp. SCC3193, phy-topathogenic fungi –Alternaria solani K-100054 (VNIIF, Bolshiye Vyazemy, Russia), Rhizoctonia solani ВКМ F-895 (VKM IBFM RAS, Pushchino, Russia).
- Results
Comments. 3.1. For better understanding and maintaining the reader's interest, the analysis from this section should have the same titles in the same order as in Materials and Methods. The authors are encouraged to check and rectify this.
Response: Done, lines 112-145
Comments. 3.2. The authors should check and revise the following expressions: 1. Total phenolic compounds per gallic acid equivalent (GAE); 2. Total flavonoids per quercetin 132 equivalent (Que); 3. Total terpenoids per linalool equivalent (Lin). They are mentioned correctly in Materials and Methods in the following lines: 590-591, 580-581. The phrase from 614-615 should be checked and corrected accordingly.
Response: Thank you for pointing this out. We agree with this comment. Done, lines 121-124
Total phenolic compounds per gallic acid equivalent (mg GAE/g DE); Total flavonoids per quercetin equivalent (mg Que/g DE); Total terpenoids per linalool equivalent (mg/g DE);
Comments. 3.3. Each Table should be revised. The authors are encouraged to register each value as a mean ± SD and mark the statistically significant values with superscripts. Moreover, in each Table footer, all abbreviations used should be explained.
Response: Done, lines 109-111 (table 1); lines 120-124 (table 2); lines 166-170 (table 3); lines 259-262 (table 4); lines 266-269 (table 5); lines 291-298 (table 6); lines 381-384 (table 7); lines 392-396 (table 7);
Comments. 3.4. For better understanding, the parameters from column 4 of Table 3 could be removed and detailed in Supplementary Material. Thus, only the main values remain, and this table's data are clearer.
Response: Done, lines 259-262 (table 4). Included Supplementary Material (Table S1)
Comments. 3.5. The authors are encouraged to display the results of antioxidant activity determinations in correspondent tables, expressed as a mean ± SD, and mark the statistically significant values. Figures 1-3, without error bars and p-values, are irrelevant.
Response: Figures 1-3 updated. (lines 226-229, 282-284, 286-288)
Comments. 3.6. The final subsection of Results could be entitled Data Analysis; the authors should analyze the correlation between all quantified constituents and the variable parameters from bioactivities determinations.
Response: Lines 249-269
- Discussion
Comments. In this section, as an overview in a final paragraph, the authors should include a great part of the Conclusions.
Response: Uptade, included
- Conclusions
In the current version, they are too long. The authors are encouraged to maintain a few briefly presented ideas that deserve to be highlighted at the final of their study. Moreover, they could mention further directions or applications.
Response: Uptade, conclusion reduced.
The presented investigation shows the results group composition and bioactivity of the extracts of juniper berries (J. communis L.) 1-and 2- years of maturity collected in the temperate continental zone of the Republic of Tatarstan, the Russian Federation use various solvents (pentane, chloroform, acetone, methanol and 70% ethanolic) and extraction methods – maceration and ultrasound-assisted maceration. The group composition of berry extracts of 1 and 2 years of maturity, obtained with the help of different extractants and conditions, differed in the quantitative composition. The identified differences in the phytopathogenic activity of extracts depended on the choice of extractants and maceration conditions. The highest antioxidant and antimi-crobial activity corresponded to acetone and 70% ethanolic extract of berries 1 year of maturity.
Further, the individual chemical composition and nematocidal activity was de-termined for 70% ethanolic and acetone extract of berries which before showed higher biological profile. A total of 38 individual compounds were identified, depending on the selected extractant (24 compounds in berries of 2nd year, 23 compounds in berries of 1st year). According to GC-MS data, the main major compounds were α-pinene (20.6-42.3%), dihydroxyacetone (0.6-39.8%), myo-inositol (4.8-19.1%), germacrene D (4.1-14.8%), and β-myrcene (3.6-8.5%). The data of nematocidal activity indicated the presence of substances toxic to soil nematodes, especially for C. elegans, in the extracts of berries of 1 year maturity, and in the presence of paraquat, the activity in-creased several times.
According to the results of this study, the 70% ethanolic and acetone extracts of berries of 1st year of J. communis L. suggesting that they can be alternatives to conven-tional synthetic pesticides and could be used in the development of biopesticide prod-ucts after additional studies necessary, including further identification of components responsible for high phytopathogenic activity.
Reviewer 3 Report
In my opinion, the objectives pursued in the present research are interesting in trying to define a chemical composition of compounds active as antioxidants and with phytopathogenic action present in Juniperus communis. Although the research seems to elucidate positive effects based on the use of different extraction methods, it is not clear to what extent the detected compounds have an effect on an individual basis. However, it is clear that the effect is of the extract obtained, and that it is not possible to state to what extent each of the components obtained have affected the results obtained - with perhaps undetected minority components being those that may be affecting the observed effects to a greater extent. The study requires extensive revision, provided that the editor considers that the above is not a handicap for publication. Broadly speaking, the objectives of the study are not clearly defined, and the introduction requires more background. Methodologically, some aspects should be clarified, and it should be explained why many extraction methods were initially used and finally only the extracts of some of them are studied. The results are not studyable in the condition shown, as no statistics are performed to compare variances, and the results of the controls are not included in many cases. The discussion should be improved by reasoning the results and including existing literature that is important and has not been mentioned. The conclusions are somewhat diffuse, as the majority components are mentioned, but it is not known to what extent each component may have affected the results (as mentioned above). In my opinion, this study requires further testing to demonstrate which of the detected components are producing the pathogenicity results obtained or whether it is produced by a summative or synergistic effect. At the writing level the manuscript requires revision by a native English speaker, and at the formatting level it needs to be improved.

English must be improved.
Author Response
Comments. Line 3: Latin name of plant in the title should be given in Italic.
Response: Thank you for pointing this out. We agree with this comment. We changed it and emphasize this point. Done, Juniperus communis L.
Comments. Line 6: abbreviation
Response: Done - Arbuzov Institute of Organic and Physical Chemistry, FRC Kazan Scientific Center of RAS,
Comments. Line 15: “Researchers” change by an impersonal form
Response: Done – “It is looking for the most effective ways to extract bioactive substances of J. communis L. berries capable of displaying the greatest range of biological activity, namely antimicrobial potential "against phytopathogens”, antioxidant and nematocidal activity.”
Comments. Line 22-23: highlight more abundant
Response: Thank you for pointing this out. We agree with this comment. We changed it and emphasize this point.
- “they contained monoterpenes, sesquiterpenes, polysaccharides, steroids, fatty acid esters, bicyclic monoterpenes.”
Comments. Line 26: introduce this technique
Response: Done, line 24 - ultrasound-assisted maceration (UAM)
Comments. Line 28: complete name
Response: Done, line 28. - Caenorhabditis elegans; Caenorhabditis briggsae
Comments. Lines 37-40: include some references
Response: Done
Comments. Lines 41-44: include some references
Response: Done, lines 46-49
Comments. Line 48: Latin name
Response: Done
Comments. Lines 52-53: references
Response: Done
Comments. Lines 54-58: paragraph above
Response: can’t be, another Reviewer comments – “The authors are encouraged to organize the data presented in the following order: 1. systematic data (Family, Genus, Species); 2. Habitat; 3. Chemical composition; 4. Bioactivities (general presentation, not only regarding those presented in the present study); 5. therapeutical considerations in traditional medicine.”
Line 64: research
Response: line 69
Comments. Lines 66-73: bibliography
Response: Done, lines 80-86
Comments. Line 75: italic
Response: Done, line 88
Comments. Lines 86-90: this part need to include to introduction
Response: Thank you for pointing this out. We agree with this comment. We changed it and emphasize this point. Done, lines 70-74
Comments. Lines 91-95: rewrite
Response: Thank you for pointing this out. We agree with this comment. We changed it and emphasize this point. Done, line 99 – “Extraction yield, pH and electrical conductivity data are shown in Table 1.”
Comments. Lines 98-99: this part to discussion
Response: Thank you for pointing this out. We agree with this comment. We changed it and emphasize this point. Done, lines 409-410
Comments. Lines 98-99: table 1, include static, dry matter (%)
Response: Thank you for pointing this out. We agree with this comment. We changed it and emphasize this point. Done, lines 109-110
Comments. Lines 101: lower?
Response: Corrected, line 104
Comments. Lines 123-126: this part to discussion
Response: Added the following information to the introduction part. Done, lines 423-425
Comments. Lines 127-130: this part to discussion
Response: Added the following information to discussion. Done, lines 414-417
Comments. Lines 131: table 2, include static
Response: Done
Comments. Lines 148-155: Not here
Response: Added the following information to discussion. Done, lines 448-453
Comments. Lines 156-157: Please include a comment explaining why you selected these extraction methods, and whether analyses were also performed on the extracts obtained by the other extraction methods
Response: Done, lines 150-152
Comments. Lines 173: Please, include the chromatogram. Please, improve the legend with all details
Response: Thank you for pointing this out. We agree with this comment. We changed it and emphasize this point. Done, lines 173-176 (table 3). Included Supplementary Material (Table S1, Figure S1)
Comments. Lines 174-175: Details of statistics are not included in the Table and Material and Methods
Response: Done, lines 874-888. Added the following chapter 4.7. Statistical Analysis
The results were summarized as the mean values ± standard deviation (SD). The data were processed using Microsoft Excel 2016 and OriginPro 9.5 (OriginLab Co., Northampton, MA, USA). Data were compared using a non-parametric Kruskal-Wallis test. The precise p-values were calculated for the pair-wise comparisons between the groups using the Mann-Whitney test. Data analysis employed IBM SPSS 23.0 statistical software (Chicago, IL, USA). The level of p < 0.05 was considered statistically signifi-cant. Pearson correlations (p < 0.05) were used to identify correlations between phyto-chemical constituents and antioxidant activity of acetone, methanol and 70% ethanolic extracts of juniper berries (J. communis L.) 1st and 2- years of maturity.
Statistical processing of the results of nematocidal activity was carried out using the Fisher angular transformation. Regression analysis of calibration characteristics and mathematical prediction of the peak area response for chromatographic analysis were performed using OriginPro 9.5 software (Origin Lab Corp, Northampton, MA, USA) and processed in accordance with the standards [93].
Comments. Lines 233: Both figures should be integrated into a single figure, including a statistical comparison including age and extraction method as factors. Controls are missing. Same comments for Figures 2 and 3
Response: updated, lines 226-229; lines 282-28, lines 286-288
Comments. Lines 277-281: Not here, I think you didn't included studies on plants concerning phytopathogenicity
Response: Done, lines 818-823. In chapter 4.6.2. Determination of antimicrobial activity. The lines include diseases that cause these pathogens
Comments. Lines 282: Legends of Tables and Figures should be improved. Please, include statistics and controls
Response: Done, lines 291-293
Comments. Table 4: I think you should improve the way to show them
Response: Done, lines 294-298
Comments. Lines 285-286: Not clear. Incomplete data
Response: Done, lines 294-298. Included control, legend, statistics
Comments. Lines 369: Same comments of previous Table
Response: updated, lines 381-384
Comments. Table 5: Controls were not included
Response: Done, lines 381-384 (table 7). Control - propylene glycol (concentrations of 0,5% and 0,025%)
Comments. Lines 378: Same comments of previous Table
Response: Done, lines 392-396 (table 8). Included control - propylene glycol (concentration of 0,5%)
Comments. Table 6: How can you explain these differences? Efficiency was increased 4-fold, but this did not occur with the rest of experimental conditions
Response: Thank you for pointing this out. We agree with this comment. Done, lines 400-403. The increase in nematodes mortality after joint action of paraquat and ethanol extracts from 1- and 2-year maturity juniper berries made it possible to suppose that these ex-tracts contain substances with prooxidant action, and in such a way they increase the toxic paraquat action on C. elegans.
Comments. Lines 385: Discussion should be improved. The results should be discussed with the data found in the bibliography and with an explanation of the results obtained. This is difficult in most cases because you haven't done any statistical analysis of comparisons
Response: Corrected. Included comparison of each group composition with the bibliographically data.
Comments. Lines 432: Please, rewrite
Response: Corrected. “Elboughdiri et al [46] found that analysis of 2nd year berry ethanol extract showed that the composition was dominated by monoterpenes (α-pinene: 39.12%, sabinene: 8.87%, β-pinene: 12.68%, myrcene: 12.92%, limonene: 2.23%) and sesquiterpenes (β-caryophyllene: 4.41%, α-humulene: 1.05%, germacrene D: 4.23%, δ-cadinene: 1.35%)”
Comments. Lines 440-441: decreased by 1.5-1.9 times in respect to
Response: update, line 493 – “Depending on the extraction method, the antioxidant activity decreased by 1.5-1.9 times with the use of ultrasound, which may be due to cavitation processes when using ultrasound and the reactions of intermolecular interaction occurring at the same time.”
Comments. Lines 464-468: This belongs to Introduction.
Response: Added the following information to the introduction part, lines 80-83
Comments. Lines 470: Brodowska et al. [43] found that ...
Response: corrected, line 510
Comments. Lines 494-496: Please, explain
Response: update, lines 544-550.
“Additionally, the behavior sensitivity to the action of acetone extract was revealed in C. elegans (96.5%), which turned out to be 46% higher than in C. briggsae (Table 7). So one might suppose, the presence of substances toxic to soil nematodes, especially for C. elegans, in berry extracts of 1st year. This was revealed in the high sensitivity of C. elegans behavior to the negative effects of J. communis L. extracts. We are suggesting that the increased activity of 1-year-old berry extracts compared to 2-year-old berry extracts is associated with a higher content of phenolic compounds in them [61, 62], monoterpenes, in particular limonene (0,502-0,546%) [63], sesquiterpenes and other biological active compounds.”
Comments. Lines 506-507: You can develop this a little more.
Response: update, line 555-559.
Extracts of other species of the genus Juniperus had weak nematocidal activity. A few articles described toxic action of Juniperus spp. against plant parasitic nematodes. Among them is the article of Kong et al. (2006) in which activity of J. communis, J. ox-ycedrus and J. virginiana essential oils against pine wood nematode Bursaphelenchus hylophilus was studied [67].
Comments. Lines 509: What was the paraquat addition based on?
Response: update, line 561 – “known as prooxidant due to its ability to generate reactive oxygen species”
Comments. Lines 516-517: Can you include a GPS sampling point
Response: Done, lines 570-571
Vysokogorsky district of Tatarstan (55.953357° N 49.183376° E).
Comments. Lines 527: Can you include a comparative photograph with berries of 2 years?
Response: Thank you for pointing this out. We agree with this comment. Done, line 584 - Figure 4. Conifers with berries of 1 year (A) and 2-year (B) of maturity J. communis L.
Comments. Lines 528: trees?
Response: Corrected
Comments. Lines 529: tree?
Response: Corrected
Comments. Lines 529: were
Response: Corrected
Comments. Lines 533: Which one's were not analytically pure?
Response: Corrected, line 587 – “All chemicals and reagents used were of analytical grade”
Comments. Lines 534-535: , and paraquat
Response: Corrected
Comments. Lines 535: were
Response: Corrected
Comments. Lines 536: acid, and quercetin were
Response: Corrected
Comments. Lines 539: it should be deleted
Response: Corrected
Comments. Lines 540: In a first stage, ...You should define point by point the main objectives of this work in the Introduction (at the end).
Response: Corrected, line 588
The first stage of the work was to obtain extracts from frozen juniper berries by maceration.
Comments. Lines 547: include company and country
Response: included - “RCT basic, IKA, Germany”
Comments. Lines 551: include also company and country. The same comment for the rest of laboratory apparatus or equipments
Response: included – “H3-18KR centrifuge, Hunan Kecheng Instrument EquipmentCo, Ltd., China”
Comments. Lines 552: According to the formatting rules it should be changed to long hyphen.
Response: Corrected
Comments. Lines 554: change it to long hyphen.
Response: Corrected
Comments. Lines 556: In a second stage, ... Did you want to compare the efficiency of both methods?
Response: Thank you for pointing this out. You're right.
Comments. Lines 563: 1% solution in water?
Response: Corrected, lines 611-613: «dried extracts were re-dissolved (to obtain a 1% solution of extracts) in a solvent eutectic consisting of 65% 1,3-propylene glycol, 25% ethanol and 10% water. »
Comments. Lines 574: mL
Response: Corrected
Comments. Lines 616-617: Can you describe it briefly as you did with the previous methods? Include also units of measurement
Response: Corrected, lines 664-676
The ascorbic acid content in extract was determined by spectrophotometric method using potassium permanganate as a chromogenic reagent at a wavelength of 530 nm [76]. Standard solutions of ascorbic acid were prepared within the samples range (1 to 100 mg/L), which dissolved (0.01 g) in small amount of 0.5 % oxalic acid solution and completed with distilled water to obtain a concentration of 100 mg/L. The solution of chromogenic agent with 100 mg/L concentration was prepared by dissolving previously 0.01 g of KMnO4 in 5 molar solution of sulfuric acid in a 100 mL volumetric flask and completed to the mark with distilled water. 1 mL of chromogenic agent was added in a series of 10 mL standard solution with different concentrations of ascorbic acid and after 5 min was performed the absorbance measurement of each solution at 530 nm against blank. The determination of ascorbic acid concentration in dry extracts was carried out in a similar way by replacing the suspension of the standard with an aliquot of the sample dissolved in water.
Comments. Lines 620: Italics. Please, change the format in all the manuscript
Response: Corrected
Comments. Lines 633: revise format.
Response: Corrected – “Whatman filter paper Grade 41”
Comments. Lines 651: I think you should delete it to avoid be confussed with a minus sign
Response: Corrected
Comments. Lines 654: include only one or two decimals and uniformize it in the rest of the document
Response: Corrected
Comments. Lines 658-660: I think you should include a new section on statistical analysis. Also include details of the rest of the statistical analysis performed
Response: Done, lines 874-888. Added the following chapter 4.7. Statistical Analysis
Comments. Lines 667-669: describe it briefly as you did with DPPH method
Response: included
The chemiluminescent assay is described in [25] and adapted to the Lum-1200 luminometer (LLC DISoft, Russia) [83]. The results were processed on a personal computer using the PowerGraph and OriginLab software. Luminol (Alfa Aesar) solution (1 mmol L–1) was prepared by dissolution of luminol in aqueous 0.1 М NaOH. Immediately before the analysis, the stock solution of luminol was diluted 4-fold with distilled water.
Chemiluminescence analysis was carried out as follows. Thermostated at 30 °С cell was charged with 1 mL of the reaction mixture containing 400 μL of 250 μМ luminol solution, 500 μL of 0.5 М of Tris buffer (Fisher Chemical), pH 8.6, and 100 μL of 40 мМ aqueous 2,2´-azobis(2-methylpropionamidine) dihydrochloride (AAPH) solution (Acros Organics). The basic chemiluminescence signal was measured for 10 min, then 10 μL of the solution of the test compound was added, and the chemiluminescence signal was acquired.
Comments. Lines 677: as
Response: Done
Comments. Lines 679-685: This should be placed in Discussion
Response: Added the following information to the introduction part, lines 484-489
Comments. Lines 688-690: Where did you obtain these strains?
Response: Done, lines 767-771
Clavibacter michiganensis gram-positive strain VKM Ac-1404 (VKM IBFM RAS, Push-chino, Russia), gram-negative strain of bacteria Erwinia carotovora spp. SCC3193, phy-topathogenic fungi –Alternaria solani K-100054 (VNIIF, Bolshiye Vyazemy, Russia), Rhizoctonia solani ВКМ F-895 (VKM IBFM RAS, Pushchino, Russia).
Comments. Lines 696-698: describe it briefly
Response: Done, lines 787-798
The experiments were planned to find antimicrobial activity, which was determined by the method of serial dilutions according to the methods described in [85, 86]. In experiments, the minimum inhibitory concentration (MIC) was determined by double se-rial dilution method. Fungistatic activity of alcoholic extract was measured by serial dilution in liquid medium. The extracts were tested at concentrations of 4.88...2500 mg/mL. A bacterial suspension or a piece of mushroom mycelium was placed in each tube with an extract of known concentration. After incubation, microbial viability was assessed visually and the minimum concentration contributing to the cessation of culture growth without killing it (minimum inhibitory concentration) was determined. To determine the minimum bactericidal and fungicidal concentrations of the extract, bacterial inoculum or fungal mycelium pieces taken from all tubes without visible growth were added to Petri dishes with agarized nutrient medium using a bacterio-logical loop. The minimum concentration at which bacteria were killed was considered the minimum bactericidal concentration, fungi the minimum fungicidal concentration.
Comments. Lines 700: ?
Response: Corrected
Comments. Lines 702: concentration?
Response: concentration – 0,09%
Comments. Lines 702: units?
Response: right
Comments. Lines 707: First time, please include also the complete name.
Response: minimum inhibitory concentration. Corrected
Comments. Lines 713: First time, please include also the complete name.
Response: minimum bactericidal concentration. Corrected
Comments. Lines 716-722: reorganise the information above
Response: Corrected
Comments. Lines 731: include also city and country
Response: Corrected
Caenorhabditis Genetics Centre ((CGC), University of Minnesota, Minneapolis, USA
Comments. Lines 732: "diameter" or include a standard symbol
Response: a diameter of 100 mm. Corrected
Comments. Lines 733: subscript. Please, revise the complete manuscript
Response: Corrected
Comments. Lines 734: Origin?
Response: Corrected, line 835- “Escherichia coli OP50, obtained from CGC”
Comments. Lines 735: age?
Response: updated – “with young adults (within the first day after L4 larva stage).”
Comments. Lines 742: min no minutes
Response: Corrected
Comments. Lines 743-745: rewrite
Response: Corrected, lines 845-847
To assess the toxicity of juniper extracts, nematodes were transferred into clean glass centrifuge tubes 10 mL in volume (50 worms in each tube). After the worms settled onto the bottom of the tubes, the supernatant was removed, and M9 buffer and juniper extracts were added.
Comments. Lines 751-752: include details in a new Statistical section
Response: Corrected. Added the following chapter 4.7. Statistical Analysis
Comments. Lines 753: delete this sub-point and merge it with 4.8.
Response: Corrected
Comments. Lines 769: highlight these new data with details
Response: Corrected
Round 2
Reviewer 1 Report
I have no further comments.
Author Response
Very grateful for your expertise in helping me improve my research article
Reviewer 2 Report
1. The reviewer appreciates the authors' efforts in reviewing their MS, according to previous comments.
2. The authors responded to 99% of comments and suggestions because the registration numbers of voucher specimens are still missing.
3. The graphs from Figures 2 and 3 are too loaded, containing numeric data, error bars, and *. The authors are encouraged to remove the numeric data for a better understanding of the graphs and to present the essential values in the MS text.
4. The authors are invited to check and mention the significance of all abbreviations in the table footer/figure caption.
5. The authors should check the entire MS text and maintain the same style of the species' scientific name (Juniperus communis L.). They mentioned the whole name in the first rows of the Introduction section. Therefore, in the rest of the MS, the authors should use the abbreviation J. communis, not J. communis L. (line 105) or Juniperus communis L. (line 116). Moreover, the reader knows from the beginning that juniper berries are from J. communis; therefore, the authors should avoid repeating them (line 110).
6. 70% ethanol is better than 70% ethanolic (as in Table 2 or line 684), ethanolic (line 1033), or 70% EtOH (Figures 1 and 2). Please check and correct the entire MS text.
7. Tables 4 and 5: Phytoconstituents or phytochemicals are better than "phytochemical c constituents". Without "group composition," the table captions are better.
8. Please check and reformulate the phrases from lines 1033-1035, 1276-1282, 1521-1524
9. As a general impression, the presentation of the MS data in the entire text is cumbersome. The authors (and, consequently, the future readers) get involved in the maturation years (first and second) of the juniper fruits, solvents, methods of obtaining the extracts (maceration and ultrasound-assisted maceration), and all data presentation is negatively affected.
The authors are encouraged to establish some abbreviations from the beginning. Thus, they can be used in the entire MS text.
For example, JB1 = Juniper berries in the first year of maturation; JB2 = Juniper berries in the second year of maturation.
All JB1 extracts in pentane, chloroform, acetone, methanol, and 70% ethanol could be abbreviated as JB1P, JB1C, JB1A, JB1M, and JB1E. The same suggestion is available for JB2, only changing the number 1 with 2.
The extracts obtaining methods could be marked as M (maceration) and US (ultrasound maceration).
Thus, for JB1, the correspondent extracts obtained by maceration would become JB1P-m, JB1C-m, JB1A-m, JB1M-m, JB1E-m; those obtained by ultrasound-assisted maceration would become JB1P-usm, JB1C-usm, JB1A-usm, JB1M-usm, JB1E-usm. Number 1 is changed with 2 for JB2.
The authors' abbreviations for ultrasound-assisted maceration as UAM (line or Mu are unsuitable because ultrasound is marked as us in all scientific papers.
All these abbreviations could be used in the entire MS text, Figures, and Tables, making the presentation more explicit and practical for the reader and highlighting the present research's value. All abbreviations should be explained in the figure caption/table footer in all Figures and Tables.
The authors should rigorously check the entire text of the current version, correcting all the misprints and verifying the structure of the long phrases for better understanding. The reviewer believes the entire MS text would become clearer and the data presentation more simple and precise using all suggested abbreviations.
The titles of several subsections should be simplified or corrected as follows:
-2.1. Phisico-chemical Properties
-2.2. Quantification Of The Main Phytochemicals
-2.3. GS-MS Analysis
-2.4. DPPH-free radical scavenging assay; "Analysis of the extracts' effect on DPPH-scavenging activity" is incorrect because DPPH-free radical scavenging activity is an effect. It is a method of antioxidant potential evaluation by measuring antiradical activity.
-2.5. Antimicrobial activity
-2.6. Nematodicidal activity ("Nematocidal" is incorrect)
The same corrections should be performed in Materials and Methods.
The reviewer suggests an English-native researcher in phytochemistry should read the entire MS text before resubmission because the current version has numerous errors. The authors appear to be unfamiliar with English scientific terms and make mistakes unintentionally.
After all these suggested corrections, the reviewer considers the MS suitable for publication in a Q1 Journal as Plants MDPI.
Author Response
Comments. The graphs from Figures 2 and 3 are too loaded, containing numeric data, error bars, and *. The authors are encouraged to remove the numeric data for a better understanding of the graphs and to present the essential values in the MS text.
Response: Thank you for pointing this out. Corrected
Comments. The authors are invited to check and mention the significance of all abbreviations in the table footer/figure caption.
Response: Thank you for pointing this out. Corrected
Comments. The authors should check the entire MS text and maintain the same style of the species' scientific name (Juniperus communis L.). They mentioned the whole name in the first rows of the Introduction section. Therefore, in the rest of the MS, the authors should use the abbreviation J. communis, not J. communis L. (line 105) or Juniperus communis L. (line 116). Moreover, the reader knows from the beginning that juniper berries are from J. communis; therefore, the authors should avoid repeating them (line 110).
Response: Thank you for pointing this out. We agree with this comment. Corrected
Comments. 70% ethanol is better than 70% ethanolic (as in Table 2 or line 684), ethanolic (line 1033), or 70% EtOH (Figures 1 and 2). Please check and correct the entire MS text.
Response: Thank you for pointing this out. We agree with this comment. Corrected
Comments. Tables 4 and 5: Phytoconstituents or phytochemicals are better than "phytochemical c constituents". Without "group composition," the table captions are better.
Response: Thank you for pointing this out. We agree with this comment. Added phytochemicals.
Comments. Please check and reformulate the phrases from lines 1033-1035, 1276-1282, 1521-1524
Response: Corrected
Comments. As a general impression, the presentation of the MS data in the entire text is cumbersome. The authors (and, consequently, the future readers) get involved in the maturation years (first and second) of the juniper fruits, solvents, methods of obtaining the extracts (maceration and ultrasound-assisted maceration), and all data presentation is negatively affected.
The authors are encouraged to establish some abbreviations from the beginning. Thus, they can be used in the entire MS text.
For example, JB1 = Juniper berries in the first year of maturation; JB2 = Juniper berries in the second year of maturation.
All JB1 extracts in pentane, chloroform, acetone, methanol, and 70% ethanol could be abbreviated as JB1P, JB1C, JB1A, JB1M, and JB1E. The same suggestion is available for JB2, only changing the number 1 with 2.
The extracts obtaining methods could be marked as M (maceration) and US (ultrasound maceration).
Thus, for JB1, the correspondent extracts obtained by maceration would become JB1P-m, JB1C-m, JB1A-m, JB1M-m, JB1E-m; those obtained by ultrasound-assisted maceration would become JB1P-usm, JB1C-usm, JB1A-usm, JB1M-usm, JB1E-usm. Number 1 is changed with 2 for JB2.
Response: Corrected, abbreviations added.
Comments The authors' abbreviations for ultrasound-assisted maceration as UAM (line or Mu are unsuitable because ultrasound is marked as us in all scientific papers.
Response: Corrected.
Comments All these abbreviations could be used in the entire MS text, Figures, and Tables, making the presentation more explicit and practical for the reader and highlighting the present research's value. All abbreviations should be explained in the figure caption/table footer in all Figures and Tables.
Response: Thank you for pointing this out. Corrected in figure / table
Comments The authors should rigorously check the entire text of the current version, correcting all the misprints and verifying the structure of the long phrases for better understanding. The reviewer believes the entire MS text would become clearer and the data presentation more simple and precise using all suggested abbreviations.
Response: Thank you for pointing this out. Corrected.
Comments The titles of several subsections should be simplified or corrected as follows:
-2.1. Phisico-chemical Properties
-2.2. Quantification Of The Main Phytochemicals
-2.3. GS-MS Analysis
-2.4. DPPH-free radical scavenging assay; "Analysis of the extracts' effect on DPPH-scavenging activity" is incorrect because DPPH-free radical scavenging activity is an effect. It is a method of antioxidant potential evaluation by measuring antiradical activity.
-2.5. Antimicrobial activity
-2.6. Nematodicidal activity ("Nematocidal" is incorrect)
The same corrections should be performed in Materials and Methods.
Response: Thank you for pointing this out. We agree with this comment. Corrected.
Reviewer 3 Report
In my opinion the manuscript has been significantly improved. However, I note here some necessary pre-publication issues:
- I have noticed some minor typographical and formatting errors. Please proofread the manuscript carefully before resubmitting it.
- Statistically it is incomplete, as there are two factors, extraction method and maturation time. Strictly a Two-Way ANOVA should be performed and indicate if there is also an interaction between the two factors. If only One-Way ANOVA is performed, both variables should be compared separately and clearly understable. In the Tables and Figures it is not clear what is being compared. Perhaps the stars could be replaced by letters to show the significant differences between the different treatments. Section 4.7 should be improved and indicate more detail of the statistical analyses performed.
- The legends of the figures and tables are incomplete; it should be possible to understand the figures by reading only the legends. Many of the abbreviations are also not included.
- Please review the numbering of the bibliography, as it should be correlative in the text and correspond to the details in the references section.
- For the purpose of improvement, details of the most important differences observed in the comparisons made could also be quantitatively indicated in the text.
In my opinion, English is fine. Although I am not a native English speaker.
Author Response
Comments 1 I have noticed some minor typographical and formatting errors. Please proofread the manuscript carefully before resubmitting it.
Response: Thank you for pointing this out. Corrected
Comments 2 Statistically it is incomplete, as there are two factors, extraction method and maturation time. Strictly a Two-Way ANOVA should be performed and indicate if there is also an interaction between the two factors. If only One-Way ANOVA is performed, both variables should be compared separately and clearly understable. In the Tables and Figures it is not clear what is being compared. Perhaps the stars could be replaced by letters to show the significant differences between the different treatments. Section 4.7 should be improved and indicate more detail of the statistical analyses performed.
Response: Thank you for pointing this out. Added, the following chapter - 2.2.1. Statistical analysis of influence of factors on extraction yield and the main phytochemicals (lines 154-166 and Supplementary Material (Table S1-S6))
Comments 3 The legends of the figures and tables are incomplete; it should be possible to understand the figures by reading only the legends. Many of the abbreviations are also not included.
Response: Thank you for pointing this out. Corrected
Comments 4 Please review the numbering of the bibliography, as it should be correlative in the text and correspond to the details in the references section.
Response: Thank you for pointing this out. Corrected
Comments 5 For the purpose of improvement, details of the most important differences observed in the comparisons made could also be quantitatively indicated in the text.
Response: Thank you for pointing this out. Corrected